# The step-wise pathway of septin hetero-octamer assembly in budding yeast

Andrew Weems, Michael McMurray*

Department of Cell and Developmental Biology, University of Colorado Anschutz Medical Campus, Aurora, United States

**Abstract** Septin proteins bind guanine nucleotides and form rod-shaped hetero-oligomers. Cells choose from a variety of available septins to assemble distinct hetero-oligomers, but the underlying mechanism was unknown. Using a new in vivo assay, we find that a stepwise assembly pathway produces the two species of budding yeast septin hetero-octamers: Cdc11/Shs1–Cdc12–Cdc3–Cdc10–Cdc10–Cdc3–Cdc12–Cdc11/Shs1. Rapid GTP hydrolysis by monomeric Cdc10 drives assembly of the core Cdc10 homodimer. The extended Cdc3 N terminus autoinhibits Cdc3 association with Cdc10 homodimers until prior Cdc3–Cdc12 interaction. Slow hydrolysis by monomeric Cdc12 and specific affinity of Cdc11 for transient Cdc12•GTP drive assembly of distinct trimers, Cdc11–Cdc12–Cdc3 or Shs1–Cdc12–Cdc3. Decreasing the cytosolic GTP:GDP ratio increases the incorporation of Shs1 vs Cdc11, which alters the curvature of filamentous septin rings. Our findings explain how GTP hydrolysis controls septin assembly, and uncover mechanisms by which cells construct defined septin complexes.

## Introduction

The septin family of cytoskeletal proteins is highly conserved between humans, yeast, and other non-plant eukaryotes, both in structure and function (*Oh and Bi, 2011*; *Mostowy and Cossart, 2012*; *Dolat et al., 2014*; *Fung et al., 2014*). Septins have been implicated in a variety of processes, including ciliogenesis, cytokinesis, microtubule dynamics, and neuronal development (*Oh and Bi, 2011*; *Mostowy and Cossart, 2012*; *Dolat et al., 2014*; *Fung et al., 2014*). Consequently, septin dysfunction is linked to a wide variety of human diseases, including cancer, male infertility, and hereditary neuralgic amyotrophy (*Oh and Bi, 2011*; *Mostowy and Cossart, 2012*; *Dolat et al., 2014*; *Fung et al., 2014*). The diversity of functions correlates with the variety of septin monomers (13 in humans, many of which have multiple distinct splice variants) that create many different oligomer species. The majority of human septins are constitutively expressed across all cell types (*Mostowy and Cossart, 2012*). It thus remains unclear how cells assemble only the oligomer species that are functionally appropriate.

Septins were first identified in *Saccharomyces cerevisiae* as temperature-sensitive mutants in which cytokinesis and bud morphogenesis are perturbed at the restrictive temperature (*Hartwell, 1971*). Consistent with functions in cell division and morphogenesis, the five septin proteins expressed in proliferating yeast cells – Cdc3, Cdc10, Cdc11, Cdc12, and Shs1 – co-localize in filamentous, ring-like structures at the mother-bud neck (*Oh and Bi, 2011*; *Glomb and Gronemeyer, 2016*). Yeast septin rings act as plasma membrane diffusion barriers and scaffolds for the recruitment of other factors (*Oh and Bi, 2011*; *Glomb and Gronemeyer, 2016*). The five yeast septin proteins co-assemble into two different species of linear, palindromic hetero-octamers that act as the basic building blocks of septin filaments (i.e. protofilaments, [*Oh and Bi, 2011*; *Glomb and Gronemeyer, 2016*]) (*Figure 1A*). The paralogs Shs1 and Cdc11 compete for occupancy of the terminal subunit positions (*McMurray et al., 2011*; *Finnigan et al., 2015*; *Garcia et al., 2011*; *Iwase et al.,*

*For correspondence: michael.
mcmurray@ucdenver.edu

**Competing interests:** The authors declare that no competing interests exist.

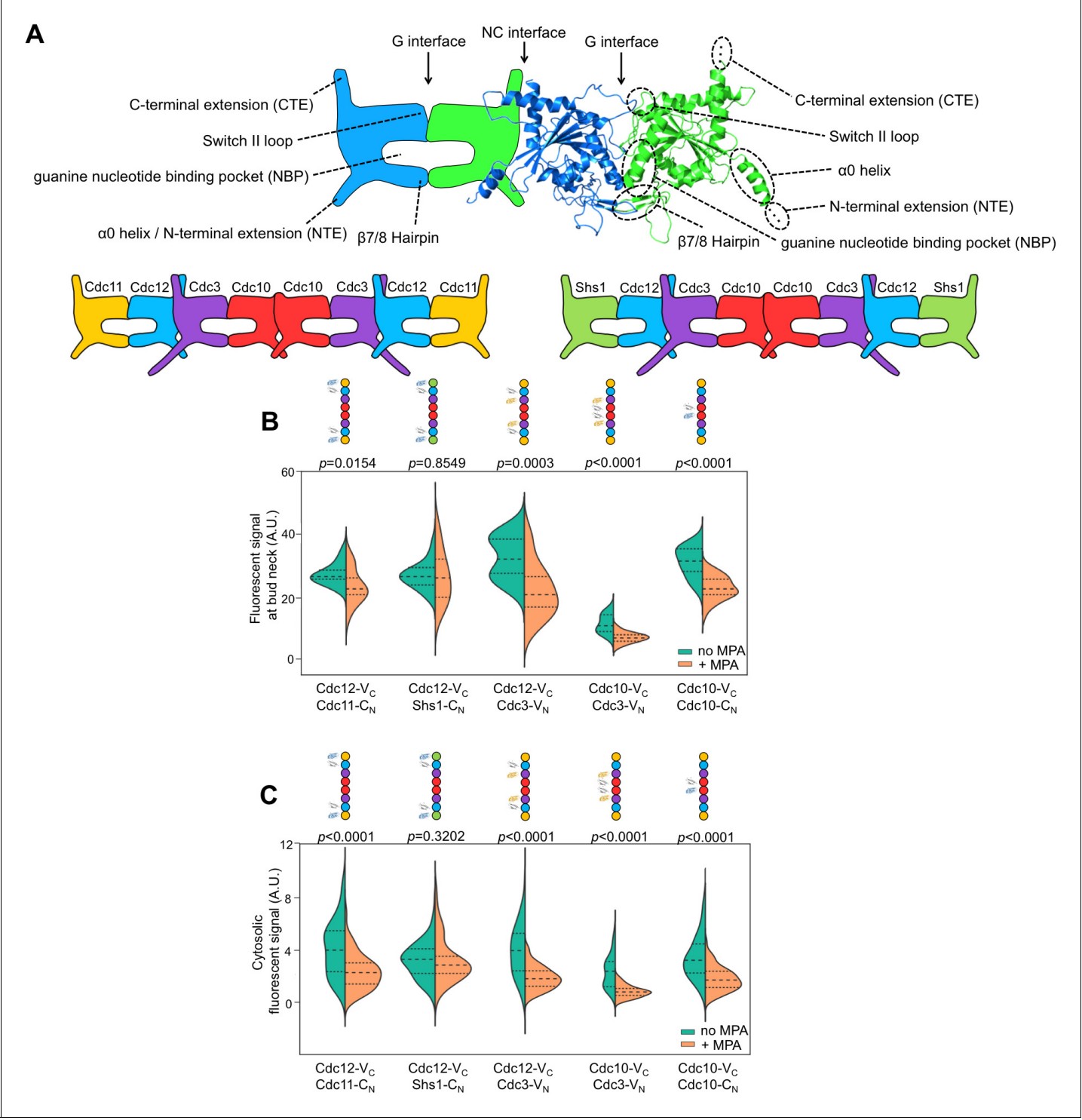

**Figure 1.** Depletion of guanine nucleotides with MPA perturbs both the G and NC septin-septin interfaces. (**A**) Diagrams illustrating septin structural elements referred to in this work, as well as the two septin protofilament species in mitotic *S. cerevisiae* cells. (**B–C**) Violin plots of BiFC signal measured at the bud necks (**B**) or in the cytosol (**C**) of cells expressing the indicated BiFC fusions, in the presence and absence of MPA. p-Values were calculated using two-tailed t-test or Mann-Whitney test, as appropriate for the distributions of the relevant data sets, which was determined by Pearson-D'Agostino normality test. Dashed lines separate quartiles. (**B**) From left to right, n = 17, 18, 26, 22, 14, 25, 10, 24, 27, and 47. (**C**) From left to right, n = 80, 54, 143, 44, 68, 79, 61, 36, 150, and 150. *Strains* used were: 12-$V_C$/11-$C_N$, 12-$V_C$/S-$C_N$, 12-$V_C$/3-$V_N$, 10-$V_C$/3-$V_N$, and 10-$V_C$/10-$C_N$.

The following figure supplements are available for figure 1:

*Figure 1 continued on next page*

*Figure 1 continued*

**Figure supplement 1.** Experimental workflow illustrating data acquisition, kernel density estimation, and violin plots.
**Figure supplement 2.** Representative micrographs for all quantitative microscopy methods used in this work.
**Figure supplement 3.** Septin-BiFC fusions used in this study do not compromise viability or crosslink filaments.

*2007*). These two protofilament species make distinct but poorly understood structural and functional contributions within filamentous septin rings (*McMurray et al., 2011*; *Finnigan et al., 2015*; *Garcia et al., 2011*; *Iwase et al., 2007*).

The role of GTPase activity in septin oligomer assembly is a long-standing, central question in the field. Generally, septin proteins possess a core GTPase domain flanked by N- and/or C-terminal extensions of various lengths (*Figure 1A*). The sequences immediately adjacent to the extensions comprise the 'NC interface', whereas the other dimerization interface ('G interface') encompasses the guanine nucleotide-binding pocket (NBP) (*Sirajuddin et al., 2007*). Hence, although when purified individually most septin subunits tested show some evidence of GTP binding, dimerization across the G interface buries bound nucleotides and precludes nucleotide exchange and access to outside factors (*Sirajuddin et al., 2007*). Moreover, rates of GTP hydrolysis by purified septins vary across a wide range, and many septins appear to lack hydrolytic capacity altogether (*Zent and Wittinghofer, 2014*). Thus, cycles of GTP binding, hydrolysis, and exchange are unlikely to control septin profilament assembly.

Instead, recent studies pointed clearly to nucleotide binding (and, for some septins, hydrolysis) as an early, one-time event that promotes the acquisition of conformations competent for protofilament assembly (*Zent and Wittinghofer, 2014*; *Nagaraj et al., 2008*; *Sirajuddin et al., 2009*; *Weems et al., 2014*; *Johnson et al., 2015*). Specifically, three lines of evidence support the idea that the identity of the nucleotide bound in the NBP influences the conformation of the NC interface in a way that dictates potential for septin-septin interactions, an example of allostery that we refer to as 'NC priming'. *Sirajuddin et al. (2009)* first observed that preventing GTP hydrolysis by the human septin SEPT2 prevented formation of well-diffracting crystals, and attributed this effect to mispositioning of the α0 helix, a key component of the NC interface in SEPT2•GDP crystals (*Sirajuddin et al., 2007*). If septin G dimerization drives GTP hydrolysis, as with many other G proteins (*Gasper et al., 2009*), then subsequent 'priming' of the NC interface might promote additional assembly steps. Using non-native human septin homodimers formed upon overexpression, *Kim et al. (2012)* found that G interface mutations prevent NC interactions, but not vice versa. Finally, *Nagaraj et al. (2008)* found that mutating the G interface of budding yeast Cdc12 perturbed both interactions with the G dimer partner, Cdc11, and the NC dimer partner, Cdc3. However, it remained a mystery why different septin subunits possess distinct N- or C-terminal extensions and display different GTPase behaviors in vitro, and if or how these differences might contribute to the assembly of distinct protofilament species.

In vivo, septin protofilament assembly is fast: no monomers or assembly intermediates are detectable in wild-type (WT) cells (*Johnson et al., 2015*; *Field et al., 1996*; *Frazier et al., 1998*; *Sellin et al., 2011*). In order to better understand the mechanisms that regulate septin protofilament assembly, we visualized the discrete steps by which individual yeast septins interact during protofilament assembly. We find that distinct domains and GTPase activities confer on the different yeast septin proteins unique properties that direct the incorporation of specific subunits in an orderly, step-wise assembly pathway that can be modulated by the cytosolic GTP:GDP ratio to meet changing cellular demands.

## Results

### Depletion of guanine nucleotides perturbs both G and NC septin-septin interfaces

If NC dimerization by yeast septins depends upon GTP hydrolysis, as predicted by the 'NC priming' model, then preventing GTP binding should perturb both NC and G dimerization. On the other hand, a reduction only in G dimerization would suggest that NC dimerization is an early, affinity-driven event that occurs independent of nucleotide state. We set out to test this prediction.

Previous studies of the role of nucleotide in yeast septin assembly mutated residues that are predicted to contact bound nucleotide and examined effects on septin-nucleotide interactions in vitro and septin-septin interactions via affinity co-purification (*Nagaraj et al., 2008*; *Sirajuddin et al., 2009*; *Versele et al., 2004*; *Versele and Thorner, 2004*; *Farkasovsky et al., 2005*). These methods are limited by the fact that the NBP itself is a major part of the G interface, and so these mutations might directly perturb septin-septin interactions in ways that do not necessarily involve roles for nucleotide state. Additionally, such mutations can promote the acquisition of non-native conformations that drive sequestration by cytosolic chaperones (*Johnson et al., 2015*). Finally, affinity purification methods involve cell lysis and washing steps after which residual interactions may not reflect authentic associations made in vivo. Indeed, purified septins are prone to promiscuous, non-native interactions (*Versele et al., 2004*). To mitigate these concerns, we designed an assay that allows us to observe WT septin interactions in vivo.

We exploited Bimolecular Fluorescence Complementation (BiFC, [*Kerppola, 2006*; *Hu and Kerppola, 2003*; *Hu et al., 2006*]) to assess septin-septin interactions, using C- or N-terminal fragments of the YFP derivative Venus (referred to here as $V_C$ and $V_N$, respectively) fused to septin C termini. Our strains also expressed an untagged copy of each septin, except for Cdc10-$V_C$–Cdc10-$V_N$, where both alleles were tagged. Viability and colony growth rates were unaffected by the tags (*Figure 1— figure supplement 3*), and elongated cells, a sensitive indicator of septin dysfunction (*McMurray, 2016*), were very rare ($\leq$2%) and similar to what is observed for non-BiFC fluorophore fusions (*Finnigan et al., 2015*; *McMurray, 2016*). We excluded such cells from our analysis. Finally, the BiFC events we analyzed did not perturb the normal dynamics of septin rings during the progression of bud growth and cytokinesis (*Figure 1—figure supplement 3*).

To test the effects of perturbing nucleotide binding by WT septins, we treated cells with 100 μg/ml mycophenolic acid (MPA), which inhibits inositol monophosphate dehydrogenase (IMPDH) activity, resulting in severe (~10 fold) decreases in cellular GDP and GTP (*Saint-Marc et al., 2009*). BiFC signal was measured after 24 hr MPA exposure, or in untreated cells prepared in the same way. At this concentration MPA slows, but does not halt, cell division, and septin ring morphology, the growth of buds, the progression of mitosis, and completion of cytokinesis are all unperturbed (*Sagot et al., 2005*; *Escobar-Henriques et al., 2001*). Thus, MPA has negligible effects on the septin-recruiting GTPase Cdc42, which in any case is thought to act on septins only following assembly (*Sadian et al., 2013*; *Okada et al., 2013*), and all available evidence suggests that, following translation (the efficiency of which is decreased by MPA (*Escobar-Henriques et al., 2001*)), the only proteins that must bind guanine nucleotides in order to promote septin protofilament assembly are the septins themselves.

Consistent with a requirement for nucleotide binding for both G and NC interfaces, MPA treatment significantly depleted BiFC signals, both at septin rings and in the cytosol, generated from interactions via either interface (*Figure 1A,B*). (See *Figure 1—figure supplement 1* for a summary of our methods and violin plots.) The notable exception was BiFC signal from Shs1–Cdc12, a G dimer, which was equivalent in untreated and treated cells. The finding that MPA does not perturb all septin-septin interactions argues against a non-specific effect of reduced translational efficiency, and indicates that Shs1 and Cdc12 may interact in a manner that is less dependent on GTP binding than is the case for the other septins (see below). The formation of functional septin rings despite a reduction in the kinetics/efficiency of protofilament assembly suggests that cells normally produce a surplus of protofilaments and is consistent with our previous mutation-based studies demonstrating that defects in nucleotide binding by septins slow (*Johnson et al., 2015*; *Schaefer et al., 2016*) but do not block (*Weems et al., 2014*) septin oligomerization. Notably, although the isolated C-terminal extensions (CTEs) of Cdc3 and Cdc12 have weak affinity for each other (*Versele et al., 2004*), the fact that depleting nucleotides reduces Cdc3–Cdc12 BiFC argues that C-terminal BiFC tags must

primarily report on authentic NC-interface-mediated interactions that are influenced allosterically by nucleotides in the NBPs. Indeed, the Cdc12 CTE alone cannot incorporate into septin filaments (*Versele et al., 2004*). Overall, our findings confirm the importance of guanine nucleotide binding for septin NC dimerization.

## Discrete steps in septin protofilament assembly

NC priming could explain the nucleotide dependence of septin-septin interactions. If so, G dimerization should precede NC interactions. In previous studies, the order of septin-septin interactions was indirectly inferred from contingency relationships between interactions perturbed by mutations or the absence of individual septin subunits (*Nagaraj et al., 2008*; *Versele et al., 2004*). We sought to determine interaction order more directly, and in vivo, using WT interfaces. Toward this end, we developed a novel assay based on multicolor BiFC (mBiFC), which involves co-expression of $V_C$ and $V_N$ with $C_N$, the N-terminal fragment of mCerulean. $V_C$ has equivalent affinity for both $V_N$ and $C_N$, which differ by only seven residues (*Hu and Kerppola, 2003*). Our new technique, Chronological Substrate Depletion BiFC (CSD-BiFC), exploits the effective irreversibility of the BiFC event (*Kerppola, 2006*; *Hu and Kerppola, 2003*; *Hu et al., 2006*) to determine the chronological order of protein interactions in a living cell (*Figure 2—figure supplement 1*). Upon co-expression of a $V_C$-tagged septin with $V_N$- and $C_N$-tagged partner septins, if one oligomerization event consistently occurs first, it precludes the $V_C$-tagged septin from engaging in subsequent BiFC events, thus dictating the intensity of the resulting fluorescent signals. If the $V_C$- and $V_N$-tagged septins interact first, then the yellow fluorescent signal will remain unchanged relative to a $V_C/V_N$-only control, while the blue signal is depleted relative to its control; if the $V_C$ and $C_N$-tagged septins interact first then the opposite trend will occur. (We confirmed experimentally that the C-terminal BiFC fusions generated equivalent signals for G and NC interfaces, and thus do not themselves bias towards a specific order [*Figure 2—figure supplement 3*]). On the other hand, mutual depletion of both signals upon co-expression of all three fusions indicates no preference in interaction order.

By interrogating each septin in the protofilament against its neighboring subunits, we determined the complete order of septin protofilament assembly, using signals at bud necks (unsurprisingly, given that signal depletion is key to this assay, cytosolic signals were too weak to quantify). NC priming predicts that G dimerization occurs first, driving GTP hydrolysis and triggering movement of the α0 helix. Interrogation of Cdc12–Cdc11 (G interface) and Cdc12–Cdc3 (NC interface) revealed that Cdc12-$V_C$ first dimerized with Cdc11-$C_N$ at the G interface before subsequent NC interaction with Cdc3-$V_N$, as evidenced by depletion of the yellow signal and maintenance of the blue signal (*Figure 2A*). This result is consistent with NC priming (*Figure 2A*).

By contrast, our CSD-BiFC data suggest that Cdc3-$V_C$ first interacts with Cdc12-$V_N$ (NC interface) before interacting with Cdc10-$C_N$ (G interface) (*Figure 2B*). This departure from NC priming was not unexpected, as Cdc3 oligomerizes robustly with Cdc12 both in yeast cells carrying Cdc10 mutations inhibiting G dimerization (*Nagaraj et al., 2008*) and during heterologous co-expression without Cdc10 in *E. coli* (*Versele et al., 2004*; *Farkasovsky et al., 2005*). Moreover, Cdc3 lacks the highly conserved Thr residue important for GTP hydrolysis in human septins (*Sirajuddin et al., 2009*) and purified Cdc3 does not hydrolyze GTP in vitro (*Versele and Thorner, 2004*). Thus, for Cdc3 NC priming is not an option, and the Cdc3 NC interface must be constitutively 'active', independent of the state of the bound nucleotide.

To our surprise, no other septin-septin interaction conformed to the NC priming model. Our data suggested that Cdc10-$V_C$ interacts first with Cdc10-$C_N$ (NC interface) before interacting with Cdc3-$V_N$ (G interface), and that Cdc12-$V_C$ interacts first with Cdc3-$V_N$ (NC interface) before interacting with Shs1-$C_N$ (G interface) (*Figure 2C,D*). These results indicate that Cdc10 NC homodimerization does not require prior G oligomerization with Cdc3 and, intriguingly, that Cdc12 NC oligomerization with Cdc3 requires G interaction when the G partner is Cdc11, but not when the G partner is Shs1. (Notably, the difference in Cdc3–Cdc12 interaction order relative to Cdc12–Cdc11/Shs1 demonstrates that, despite the expected proximity of the C-terminal BiFC tags on Cdc3 and Cdc12 within a protofilament, Cdc3–Cdc12 NC association must precede stable interactions between the CTEs.) We conclude that mechanisms other than NC priming drive the majority of septin-septin interactions. We note that, given the slow maturation of the reconstituted fluorophore following a BiFC event (~50 min [*Kerppola, 2008*]), compounded with long septin half-lives and septin 'recycling' through cell divisions (*McMurray and Thorner, 2008*), the BiFC signals we measured reported

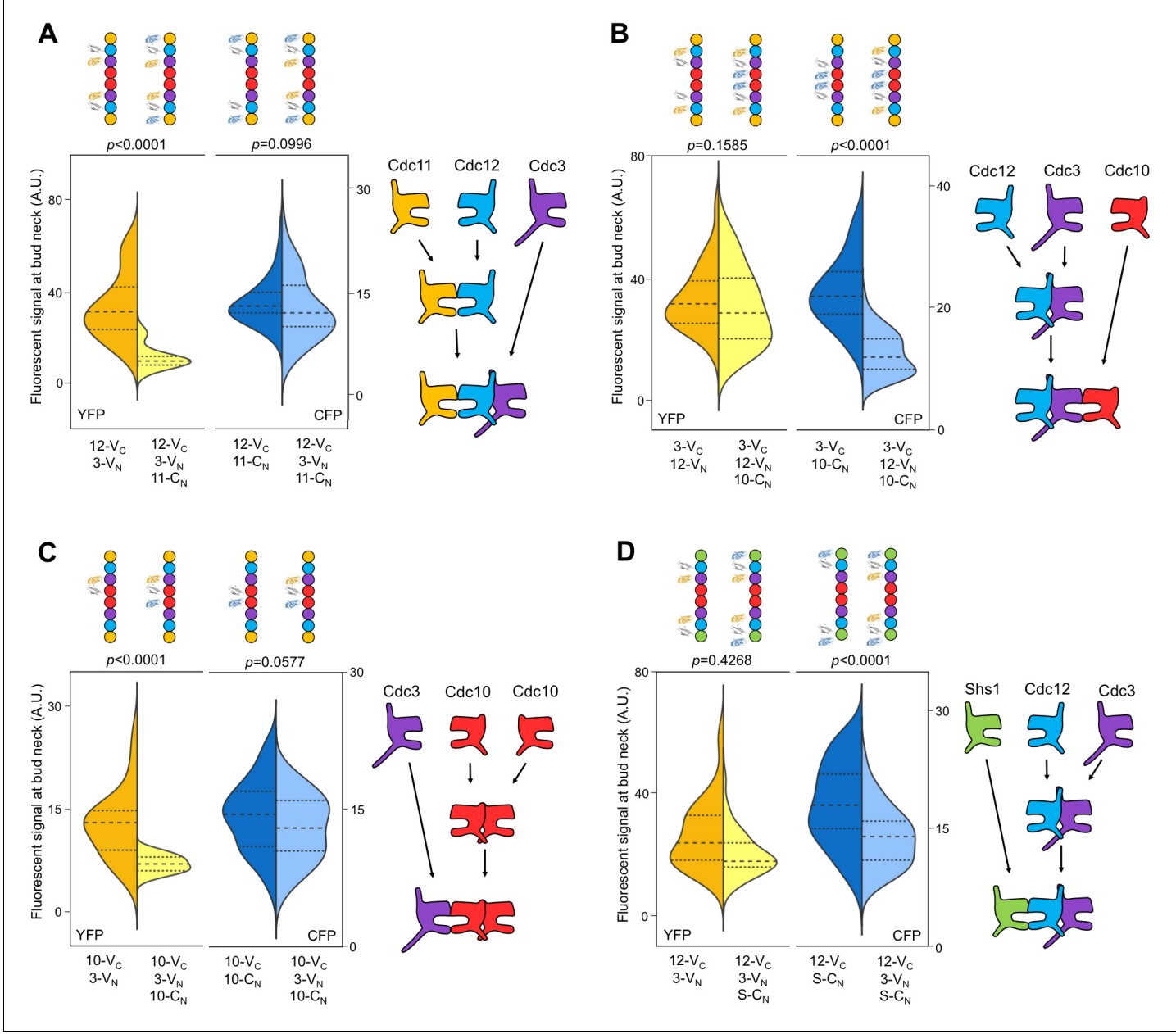

**Figure 2.** The order of septin-septin interactions during de novo assembly, according to CSD-BiFC. (A–D) As in *Figure 1B*, but for untreated cells expressing the indicated BiFC fusions, and showing YFP or CFP fluorescence. '3', Cdc3; '10', Cdc10; '11', Cdc11; '12', Cdc12; 'S', Shs1. Diagrams provide the interpretation of the results in cartoon form. (A) From left to right, $n = 40, 38, 57,$ and 45. (B) From left to right, $n = 70, 64, 54,$ and 54. (C) From left to right, $n = 26, 21, 60,$ and 39. (D) From left to right, $n = 42, 41, 78,$ and 40. Strains used were: 12-$V_C$/3-$V_N$, 12-$V_C$/11-$C_N$, 12-$V_C$/3-$V_N$/11$C_N$, 3-$V_C$/12-$V_N$, 3-$V_C$/10-$C_N$, 3-$V_C$/12-$V_N$/10-$C_N$, 10-$V_C$/3-$V_N$, 10-$V_C$/10-$C_N$, 12-$V_C$/S-$C_N$, and 12$V_C$/3-$V_N$/S-$C_N$.

The following figure supplements are available for figure 2:

**Figure supplement 1.** Conceptual layout of a CSD-BiFC experiment.

**Figure supplement 2.** The complete pathway of budding yeast septin protofilament assembly, according to CSD-BiFC.

**Figure supplement 3.** C-terminal septin-BiFC fusions do not bias toward NC interactions.

primarily on interactions that occurred during previous cell cycles. However, if even a significant fraction of septin-septin CSD-BiFC signals reflected associations between pre-formed protofilaments that occurred during polymerization at the plasma membrane, then we would expect mutual depletion in every case (i.e. apparently random order), at least according to available data for how protofilaments encounter one another during polymerization (*Bridges et al., 2014*). Thus, while these results deviate from a 'G-Dimerization First' model, our CSD-mBiFC results clearly indicate an in vivo protofilament assembly pathway that progresses not through random protein-protein interactions, but rather through specific, discrete steps. Crucially, our findings also reveal a key difference between incorporation of Shs1 vs Cdc11 during protofilament assembly, suggesting a new mechanism to generate subunit diversity.

## When Cdc10 cannot bind nucleotide, Cdc10 homodimerization no longer precedes hetero-dimerization with Cdc3

To explain our observations that NC-dimerization preceded G oligomerization for Cdc10–Cdc10–Cdc3 and Cdc3–Cdc12–Shs1, we considered an alternative possible relationship between GTP hydrolysis and the conformation of the NC interface. If Cdc12 and Cdc10 hydrolyze GTP as monomers, prior to interaction with any other septin, these proteins could effectively 'auto-prime' their NC interfaces, allowing NC dimerization to precede G dimerization. In support of this idea, monomeric Cdc10 and Cdc12 (but not Cdc11 or Cdc3) are capable of GTP hydrolysis in vitro (*Versele and Thorner, 2004*).

The NC autopriming model predicts that the ability of Cdc10 to efficiently NC homodimerize prior to G interaction with Cdc3 requires GTP binding and hydrolysis by Cdc10, and that disrupting nucleotide binding should disrupt this order of interaction. To this end, we used a substitution in the Cdc10 NBP, D182N, that prevents binding of guanine nucleotides (*Weems et al., 2014*; *Kuo et al., 2012*), and queried the order of Cdc10(D182N)-$V_C$ interaction with Cdc3-$V_N$ and/or Cdc10-$C_N$. As predicted, yellow fluorescence due to Cdc10(D182N)-$V_C$–Cdc3-$V_N$ interaction was depleted by co-expression of Cdc10-$C_N$, and blue fluorescence from Cdc10-$C_N$–Cdc10-$V_C$ interaction was depleted by co-expression of Cdc3-$V_N$ (*Figure 3*). Thus, neither septin-septin interaction consistently precedes the other, arguing that without NC-priming ability, Cdc10(D182N) is equally likely to interact first with either Cdc3 (G interface) or WT Cdc10 (NC interface). Notably, disruption of interaction order by a substitution in the Cdc10 NBP demonstrates that any added NC interface affinity due to the proximity of C-terminal BiFC tags does not introduce a bias in interaction order. When expressed as the sole source of Cdc10, the *cdc10(D182N)* allele provides largely normal function at low temperatures (*Hartwell, 1971*; *Weems et al., 2014*; *Johnson et al., 2015*), demonstrating that nucleotide binding and hydrolysis by Cdc10 are not absolutely required for septin hetero-octamer or filament assembly. Thus, when unable to rely on GTPase activity to control via NC autopriming the conformation of the NC interface, at permissive temperatures Cdc10(D182N) presumably achieves an 'active' NC conformation in a manner that is spontaneous and haphazard with regard to when Cdc10 interacts with Cdc3, reflecting a breakdown of allosteric control.

## Incorporation of Cdc11 vs Shs1 is controlled by the nucleotide state of Cdc12

Our finding that Cdc12–Cdc11 G dimerization precedes Cdc12–Cdc3 NC interaction was consistent with the NC priming model for interaction with Cdc3, but our finding that Cdc12–Cdc3 interaction precedes Cdc12–Shs1 (G interface) interaction was indicative instead of NC auto-priming. To resolve this paradox, we considered another alternative mechanism. The rate of GTP hydrolysis by purified Cdc12 is approximately half that of Cdc10 (*Versele and Thorner, 2004*). If monomeric Cdc12 GTP hydrolysis is also comparatively slow in vivo, then some Cdc12•GTP molecules might persist transiently in the cytosol before conversion to Cdc12•GDP. If Cdc12•GTP has a higher affinity for Cdc11 than for Shs1, and the reverse is true for Cdc12•GDP, then Cdc11 would associate with Cdc12 before it autoprimes its NC interface, and Shs1 would associate after NC auto-priming.

Additionally, our observation that Cdc3 consistently interacts with Cdc12 before Cdc12–Shs1 association suggests that Cdc12•GDP has a higher affinity (slower off-rate) for NC interaction with Cdc3 than for G interaction with Shs1. Similarly, Cdc10 prefers to NC homodimerize prior to interacting with Cdc3. Studies of non-septin homo-oligomers indicate that larger interfaces tend to

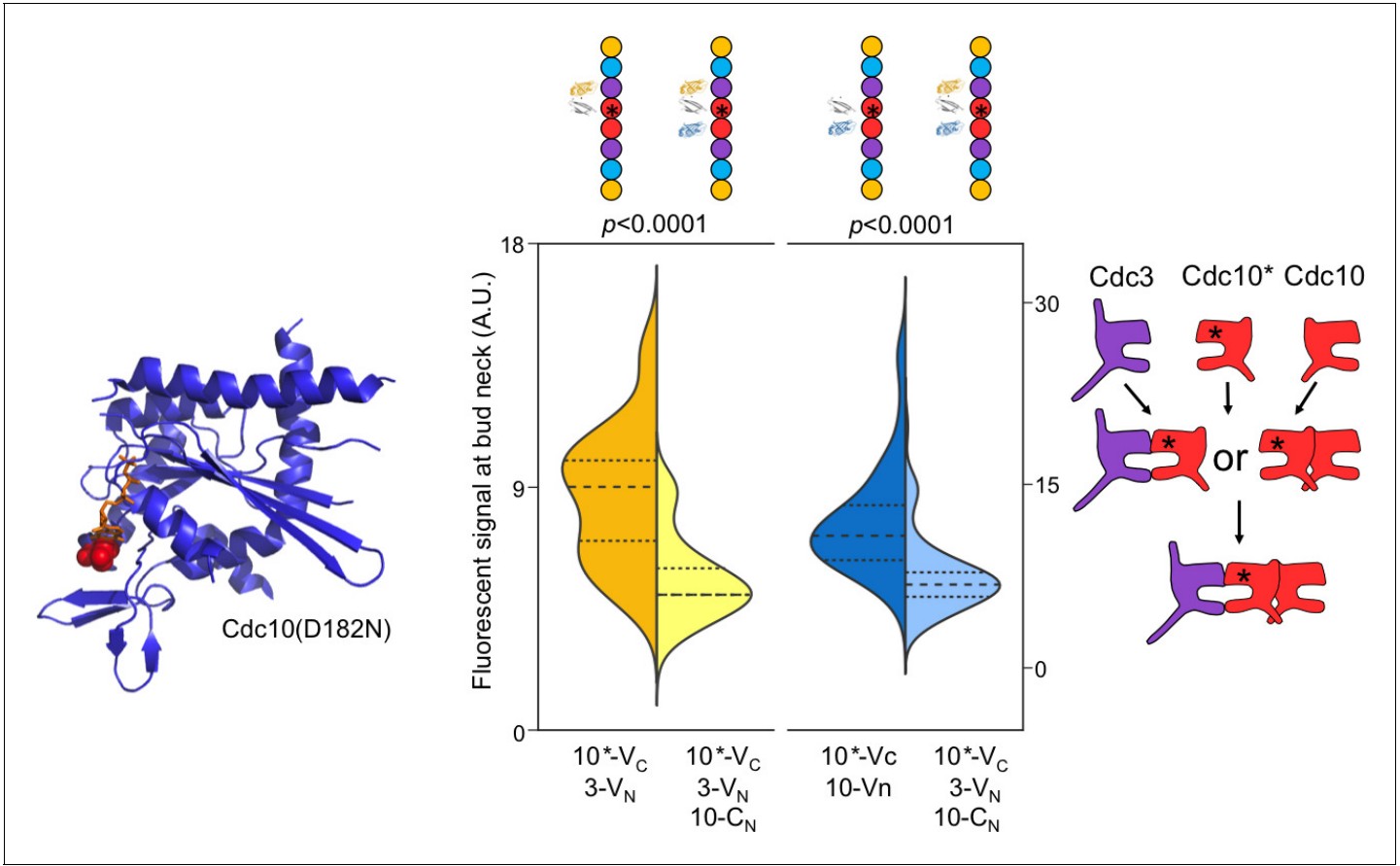

**Figure 3.** Perturbing GTP binding by Cdc10 disrupts the order of interactions with Cdc3. Left, ribbon diagram of Sept2-GppNHp (PDB 3FTQ) models the location of the mutated Asp residue (red spheres) and bound nucleotide (orange). Center, violin plot of CSD-BiFC results, as in *Figure 2C* but with Cdc10(D182N)-$V_C$ ('10*-$V_C$', 'Cdc10*'). Right, cartoon of inferred order of assembly. From left to right, $n$ = 60, 51, 95, and 62. Strains used were: 10 (D182N)-$V_C$/3-$V_N$, 10(D182N)/10-$C_N$, and 10(D182N)-$V_C$/3-$V_N$/10$C_N$.

interact first (*Levy et al., 2008*), and in crystal structures the septin NC interface buries ~60% more surface (*Sirajuddin et al., 2007*). Hence, once 'primed', septin NC interfaces in general may dissociate slower than G interfaces.

Truncation of the Cdc12 CTE accelerates GTP hydrolysis in vitro (*Versele and Thorner, 2004*), providing a genetic approach to test our model, according to the following logic. If slow GTP hydrolysis by Cdc12 is critical for creating a transient population of Cdc12•GTP monomers that preferentially recruit Cdc11, and the Cdc12ΔCTE mutant hydrolyzes GTP more rapidly, then recruitment of Cdc11 should decrease, accompanied by a corresponding increase in recruitment of Shs1. The Cdc12ΔCTE mutant is lethal as the sole source of Cdc12 (*Versele et al., 2004*) and is dominantly lethal when highly overexpressed in WT cells (*Johnson et al., 2015*; *Versele et al., 2004*). We therefore quantified the incorporation of Cdc11-GFP or Shs1-GFP into septin rings (normalized to Cdc10-mCherry) in *CDC12*[+] cells moderately overexpressing Cdc12ΔCTE. In agreement with our prediction, in cells with a Cdc12ΔCTE plasmid, Shs1 incorporation was significantly increased compared to cells with a WT Cdc12 plasmid, and Cdc11 was correspondingly depleted (*Figure 4C*).

To corroborate these results using an alternate approach, we assayed Cdc12–Shs1/Cdc11 interactions via BiFC. Cdc12ΔCTE-$V_C$–Cdc11-$V_N$ signal was reduced compared to Cdc12-$V_C$–Cdc11-$V_N$ controls, and Cdc12ΔCTE-$V_C$–Shs1-$V_N$ was elevated compared to Cdc12-$V_C$–Shs1-$V_N$ (*Figure 4A*). Thus, mutating sequences near the Cdc12–Cdc3 NC interface controls selection of the G partner of Cdc12, consistent with CTE regulation of Cdc12 hydrolysis of GTP, and specificity of Cdc11/Shs1 interaction with Cdc12 in distinct nucleotide states.

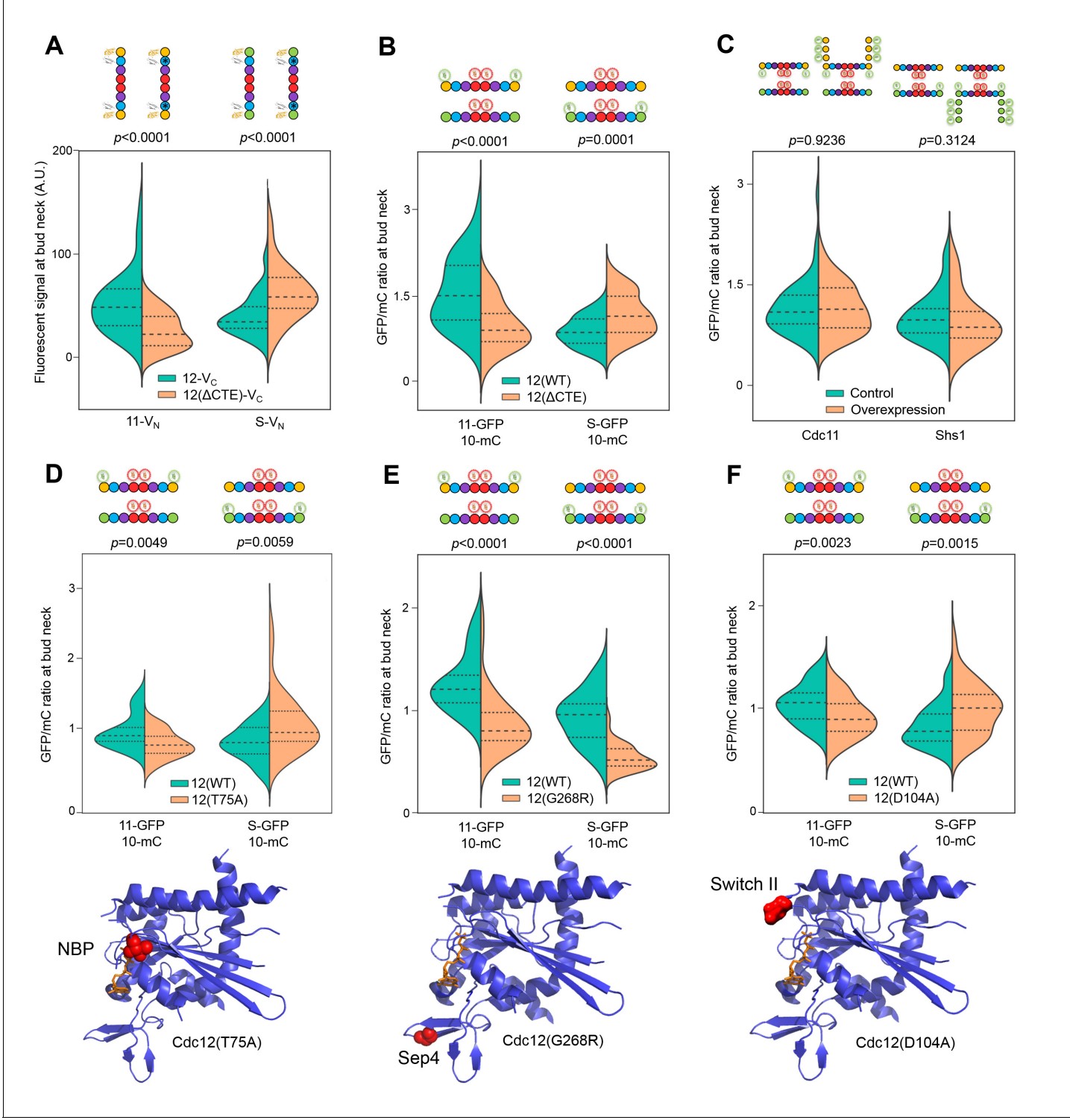

**Figure 4.** Cdc12 nucleotide state dictates the specificity of Cdc11/Shs1 recruitment via the Cdc12 Switch II loop. (**A**) Violin plot of BiFC signal at the bud necks of cells expressing Cdc11-$V_N$ or Shs1-$V_N$ (strains YEF5691 or YEF5693) and carrying a plasmid encoding Cdc12-$V_C$ (YCpUK-Cdc12-$V_C$) or Cdc12($\Delta$CTE)-$V_C$ (YCpUK-Cdc12($\Delta$CTE)-$V_C$). From left to right, $n$ = 50, 31, 47, and 27. (**B–F**) Violin plots of Cdc11-GFP or Shs1-GFP signal at bud necks expressed as a ratio over Cdc10mCherry ('mC') bud neck signal. (**B**) Cdc11-GFP/Cdc10-mC (strain JTY5396) and Shs1GFP/Cdc10-mC (strain JTY5397) cells over-expressing from high-copy plasmids either WT Cdc12 (pMVB49) or Cdc12($\Delta$CTE) (pMVB54). From left to right, $n$ = 38, 24, 40, and 42. (**C**) 'Cdc11', Cdc11-GFP/Cdc10-mCherry strain JTY5396 carrying either an empty vector plasmid (pRS316, 'Control') or a Cdc11-GFP plasmid (pSB5, 'Overexpression')."Shs1', Shs1GFP/Cdc10-mCherry strain JTY5397 carrying either an empty vector plasmid (pRS316, 'Control') or an Shs1-GFP plasmid

*Figure 4 continued on next page*

*Figure 4 continued*

(pRS316-Shs1-GFP, 'Overexpression'). (D) Cdc11-GFP/Cdc10-mC (strain JTY5396) and Shs1GFP/Cdc10-mCherry (strain JTY5397) cells overexpressing from plasmids either WT Cdc12 (pFM650) or Cdc12(T75A) (pFM829). From left to right, n = 20, 23, 28, and 32. (E) Cdc11-GFP/Cdc10-mCherry or Shs1-GFP/Cdc10-mCherry cells carrying either the WT or *cdc12(G268R)* mutant allele at the *CDC12* locus (strains used were diploids derived from mating JTY5396 or JTY5397 to the mutant strain JPTA1435 [G268R], or BY4742 as control). From left to right, n = 42, 39, 23, and 52. (F) Cdc11-GFP/Cdc10-mCherry or Shs1-GFP/Cdc10-mCherry cells carrying either WT or D104A Cdc12 plasmids (strains used were JTY5396 and JTY5397, carrying either pFM650 or YCpL-Cdc12(D104A)). From left to right, n = 39, 48, 52, and 44. Ribbon diagrams below (D–F) illustrate the locations of the mutations, as in *Figure 3*.

The following figure supplements are available for figure 4:

**Figure supplement 1.** The Cdc12 oligomerization pathway is coordinated by a GTP hydrolysis timer.

**Figure supplement 2.** Cdc11-GFP and Shs1-GFP overexpression increases fluorescent signal in the cytosol.

To further test the reliance of Cdc11–Cdc12 interaction on Cdc12 nucleotide state, we mutated Cdc12 to alter nucleotide occupancy in the NBP. Thr75 corresponds to a Thr residue that in human septins contacts the γ phosphate of GTP and is thought to catalyze GTP hydrolysis (*Sirajuddin et al., 2009*); substitution to Gly or Ala inhibits both stable binding of GTP and its hydrolysis without reducing affinity for GDP (*Sirajuddin et al., 2009*; *Abbey et al., 2016*). Consistent with a predicted depletion of Cdc12•GTP, in cells expressing *cdc12(T75A)* Cdc11 incorporation into neck filaments was significantly reduced, and Shs1 was correspondingly increased (*Figure 4D*). By contrast, Cdc11 and Shs1 were both depleted from neck filaments in cells expressing Cdc12(G268R) (*Figure 4E*), a mutant with a substitution in a key component of the G interface (the Sep4 motif) that lies outside the NBP and does not change conformation upon GTP hydrolysis (*Sirajuddin et al., 2009*). Interestingly, although levels of both Cdc11 and Shs1 in septin rings were reduced in *cdc12(G268R)* cells, the relative ratio between the two proteins (Cdc11:Shs1 ≅ 1.6:1) was preserved. These observations indicate that Shs1 is able to bind Cdc12 regardless of the nature of the nucleotide bound in the Cdc12 NBP, whereas Cdc11 is specific for a transient Cdc12•GTP species. Independence of Cdc12 nucleotide binding for Shs1–Cdc12 interaction may explain our finding that this septin-septin association was least affected by MPA treatment (*Figure 1B,C*).

## The Cdc12 switch II loop mediates communication of Cdc12 nucleotide state across the G interface

In septins, as in other small GTPases, GTP hydrolysis triggers movements of the Switch loops (I and II). Switch II in particular makes specific contacts in *trans* across the G interface, and we (*Weems et al., 2014*) and others (*Zent and Wittinghofer, 2014*) have previously demonstrated that Switch II mutations alter specificity for the nucleotide state of the G dimer partner. Thus, the Cdc12 Switch II is an excellent candidate to communicate Cdc12 nucleotide state to the future G partner, Cdc11 or Shs1. This model predicts that Cdc12 Switch II mutation should perturb the ability of Cdc12•GTP to recruit Cdc11. We mutated in Cdc12 a highly conserved Asp residue within the Switch II that is known in other septins to make *trans* contacts across the G interface (human SEPT2•GppNHp homodimer [*Sirajuddin et al., 2009*]) and to dictate G partner specificity (Cdc3–Cdc10 G heterodimer [*Weems et al., 2014*]). In support of our model, Cdc11 incorporation into septin rings decreased in *cdc12(D104A)* cells, and Shs1 incorporation increased (*Figure 4F*). These findings provide strong support for specific recruitment of Cdc11 by Cdc12•GTP via interactions made by the Cdc12 Switch II loop.

## Relative Cdc11-to-Shs1 expression levels do not dictate the ratio of Cdc11:Shs1 incorporation into higher order septin structures

According to our 'hydrolysis timer' model (*Figure 4—figure supplement 1*), the ratio of Cdc11-flanked to Shs1-flanked protofilaments is controlled primarily by the rate at which Cdc12 hydrolyzes GTP. The G interface of nascent, monomeric Cdc12•GTP binds preferentially to Cdc11, but if Cdc12 hydrolyzes GTP prior to Cdc12–Cdc11 interaction, Cdc12•GDP is no longer competent to bind Cdc11, and Shs1 instead occupies the Cdc12 G interface. This model predicts that the relative

abundance of monomeric Cdc11 and Shs1 available during protofilament assembly (Cdc11:Shs1 ratio measurements range from 0.8:1 to 1.2:1 [*Newman et al., 2006*; *Kulak et al., 2014*; *Chong et al., 2015*]) is largely irrelevant, because incorporation of each subunit is dictated by the limited availability of Cdc12 in specific nucleotide states. To test this prediction, we experimentally altered the abundance of Cdc11 or Shs1 and measured incorporation into septin rings. High-level overexpression of Cdc11 or Shs1 is toxic (*Iwase et al., 2007*; *Sopko et al., 2006*). We introduced into strains expressing Cdc11-GFP or Shs1-GFP from the chromosomal locus low-copy plasmids encoding an additional source of Cdc11-GFP or Shs1-GFP, which should increase abundance of the GFP-tagged septin two- to fourfold. Indeed, cytosolic fluorescence increased significantly (*Figure 4—figure supplement 2*). Crucially, however, there was no discernable difference in the GFP signal in septin rings (*Figure 4C*). Thus, the preferential incorporation of Cdc11 vs Shs1 into septin protofilaments cannot be explained by a higher overall Cdc12 affinity for Cdc11 or transcriptional control of Cdc11:Shs1 stoichiometry, and according to our results is best explained by specific affinity between Cdc11 and Cdc12•GTP.

## Allosteric control of Cdc3 interaction order by the Cdc3 N-terminal extension

Our CSD-BiFC experiments demonstrated that Cdc3 usually associates with Cdc12 via the NC interface before interacting via the G interface with Cdc10 (*Figure 2B*), suggesting a possible 'G priming' mechanism for Cdc3. Cdc3 G priming might represent a reversal of NC priming, in which the act of NC association with Cdc12 properly positions the Cdc3 α0 helix – via a kind of 'induced fit' – and triggers allosteric changes at the G interface that allow interaction with Cdc10. However, since Cdc3 apparently lacks the ability to hydrolyze GTP, we reasoned that Cdc3 would require some unique feature(s) to allow hydrolysis-independent G priming.

Cdc3 is unique among mitotic yeast septins in possessing a very long N-terminal extension (NTE), representing 100 residues of additional sequence N-terminal of the α0 helix. The Cdc3 NTE lacks any known motif or region of predicted secondary structure. If the NTE has some affinity for the Cdc3 G interface, then this long, flexible sequence could occlude the G interface in cis until Cdc3 interacts with Cdc12 across the NC interface, whereupon repositioning of the Cdc3 α0 helix toward Cdc12 might drive rearrangement of the Cdc3 NTE, exposing the G interface for association with Cdc10. Indeed, in the atomic structure of the equivalent interface in the human protofilament, the NC interface between SEPT7•GDP and SEPT6•GTP, a portion (~7 aa) of the short NTE of SEPT6 (~25 aa) is resolved and fits into a groove on the underside of SEPT7, pointing directly away from the SEPT6 G interface (*Sirajuddin et al., 2007*). The Cdc3–Cdc12 NC interaction would thus be predicted to position the Cdc3 NTE away from the Cdc3 G interface. Similar autoinhibitory mechanisms operate in many other proteins (*Trudeau et al., 2013*; *Pufall and Graves, 2002*): long, disordered regions occlude active sites while the protein is monomeric, and the inhibition is relieved upon dimerization and recruitment of the auto-inhibitory region to the binding partner.

The Cdc3 NTE is predicted to be almost entirely disordered (*Figure 5—figure supplement 2A*). Sequence alignments with other fungal species revealed considerable drift in the NTEs of otherwise closely related Cdc3 homologs, yet the disordered character was uniformly maintained (*Figure 5—figure supplement 2B*). These closely related NTEs contain only a small stretch of sequence that is conserved (*Figure 5—figure supplement 2B,D*); by comparison, in fungal species in which Cdc3 lacks an NTE, this conserved GQ(V/K)(L/I)PxQP sequence, just N-terminal of the α0 helix, is also missing (*Figure 5—figure supplement 2C*). This sequence may play an important role in NTE autoinhibitory functions, perhaps promoting proper positioning of the disordered region, or potentiating recruitment by Cdc12.

To look for evidence that the NTE transiently occupies the Cdc3 G interface prior to Cdc3 interaction with other septins, we considered that for Cdc10, Cdc11, and Cdc12, single NBP mutations are sufficient to slow the maturation of the mutant septin to a conformation capable of G dimerization with other septins (*Nagaraj et al., 2008*; *Johnson et al., 2015*). This delay is due in part to prolonged interactions with cytosolic chaperones that recognize misfolded septin G interfaces and is manifested in vivo as the inability of the mutant septin to incorporate into septin rings when co-expressed in haploid cells with a WT allele of the same septin (*Nagaraj et al., 2008*; *Johnson et al., 2015*). Yet for Cdc3 only a triple NBP mutation (G129V K132E T133N) sufficed to render the mutant less able to 'compete' with WT Cdc3 for incorporation into higher-order septin assemblies

(*Nagaraj et al., 2008*). These observations could be explained if the Cdc3 NTE shields a singly-substituted NBP from chaperone recognition and sequestration, consistent with our model that the NTE also effectively shields the WT Cdc3 G interface (of which the NBP is a major part) from interaction with Cdc10.

To test this theory, we introduced a single substitution in the Cdc3 NBP, Cdc3(D289N), predicted to, like Cdc10(D182N), prevent binding of guanosine nucleotides, and examined the ability of Cdc3 (D289N)-GFP to incorporate into septin rings when co-expressed with untagged, WT Cdc3. Unlike Cdc10(D182N)-GFP, which is sequestered in the cytosol (*Johnson et al., 2015*), Cdc3(D289N)-GFP was incorporated efficiently into septin rings, equivalent to the behavior of WT Cdc3-GFP tested in the same way (*Figure 5A*). Consistent with our hypothesis, the NTE was required for efficient higher-order incorporation by Cdc3(D289N) in the presence of WT Cdc3, as Cdc3(Δ1–100 D289N)-GFP was unable to localize to the septin ring in WT cells (*Figure 5B*). Even a partially truncated NTE was able to shield Cdc3(D289N) from cytosolic sequestration, as Cdc3(Δ1–56 D289N)-GFP localized robustly to septin rings (*Figure 5B*). Importantly, however, NTE truncation did not itself compromise higher order assembly by Cdc3, because Cdc3(Δ1–100)-GFP incorporated normally into septin rings (*Figure 5C*). Finally, the Cdc3(Δ1–100 D289N) mutant protein was not simply incapable of higher order assembly, because when expressed as the sole source of Cdc3 this allele provided normal septin function under the same conditions in which the co-expression experiments were performed (*Figure 5C*). Like *cdc10(D182N)* cells, but unlike cells in which either the Cdc3 NBP or the NTE was intact, *cdc3(Δ1–100 D289N)* cells were unable to proliferate at 30°C (*Figure 5C*). These results suggest that the Cdc3 NTE possesses a domain between residues 57 and 100 that is capable of recognizing and occluding a nucleotide-free Cdc3 NBP, in agreement with our model in which the NTE acts as an auto-inhibitory element to dictate the order of Cdc3 interactions with other septins.

To further explore the plausibility of Cdc3 G interface occlusion by the NTE, we generated an atomic model of monomeric nucleotide-free Cdc3 by sequence threading through the structure of monomeric nucleotide-free Cdc11 (PDB 3FTQ). We performed docking simulations between this structure and the Cdc3 NTE. Residues 57–100 suffice for shielding of Cdc3(D289N) (*Figure 5B*), so we eliminated NTE residues 1–56. We further reasoned that of the 43 functionally relevant amino acids, a linker region of 15–20 residues would be necessary to bridge the gap between the α0 helix and the NBP, so we also eliminated residues 82–100. Using Autodock Vina, we docked NTE(57-81) to our Cdc3 model, allowing full conformational freedom, and obtained a solution in which the NBP is occluded and the docked NTE region is oriented appropriately to connect via the linker with the α0 helix (*Figure 5—figure supplement 3*). (No solution could be obtained using a model containing bound nucleotide.) We believe this structure provides a plausible model for occupancy of the Cdc3 NBP by the NTE, supporting our hypothesis that the NTE shields the nascent NBP from intermolecular interactions until Cdc3–Cdc12 association has occurred.

Our model predicts that NTE deletion will disrupt the order of Cdc3 interactions with other septins. Specifically, prematurely exposing the Cdc3 G interface to Cdc10 prior to Cdc3 interaction with Cdc12 might reverse the order of interactions between Cdc3, Cdc12, and Cdc10. We generated a Cdc3(Δ1–100)-$V_C$ fusion and used it to perform CSD-BiFC with Cdc12-$V_N$ and Cdc10-$C_N$. As predicted, expression of Cdc10-$C_N$ depleted the signal arising from Cdc3(Δ1–100)-$V_C$–Cdc12-$V_N$ interaction, whereas Cdc3(Δ1–100)-$V_C$–Cdc10-$C_N$ signal was unaffected by expression of Cdc12-$V_N$ (*Figure 5D,E*). These findings demonstrate a requirement for the Cdc3 NTE in enforcing oligomerization order, specifically in preventing G-interface-mediated oligomerization with Cdc10 until after NC-interface-mediated oligomerization with Cdc12.

## Cytosolic guanine nucleotide ratios bias incorporation of Shs1 vs Cdc11

While the cytosolic ratio of GTP to GDP is often assumed to be fixed at a high value, in fact this ratio is subject to large variations under different cellular conditions, both in budding yeast (*Saint-Marc et al., 2009*; *Rudoni et al., 2001*) and human cells (*Meshkini, 2014*). Septin proteins capable of GTP hydrolysis must, by definition, be able to bind both GTP and GDP, and indeed the prediction that septins are capable of binding both free GTP and GDP is widely supported in the literature (*Sirajuddin et al., 2009*; *Farkasovsky et al., 2005*; *Zeraik et al., 2014*). We reasoned that, as the cytosolic GTP:GDP ratio approaches 1, it becomes more likely that nascent monomeric Cdc12 and Cdc10 first bind GDP rather than GTP, thus bypassing any 'NC auto-priming' step. For Cdc12, one

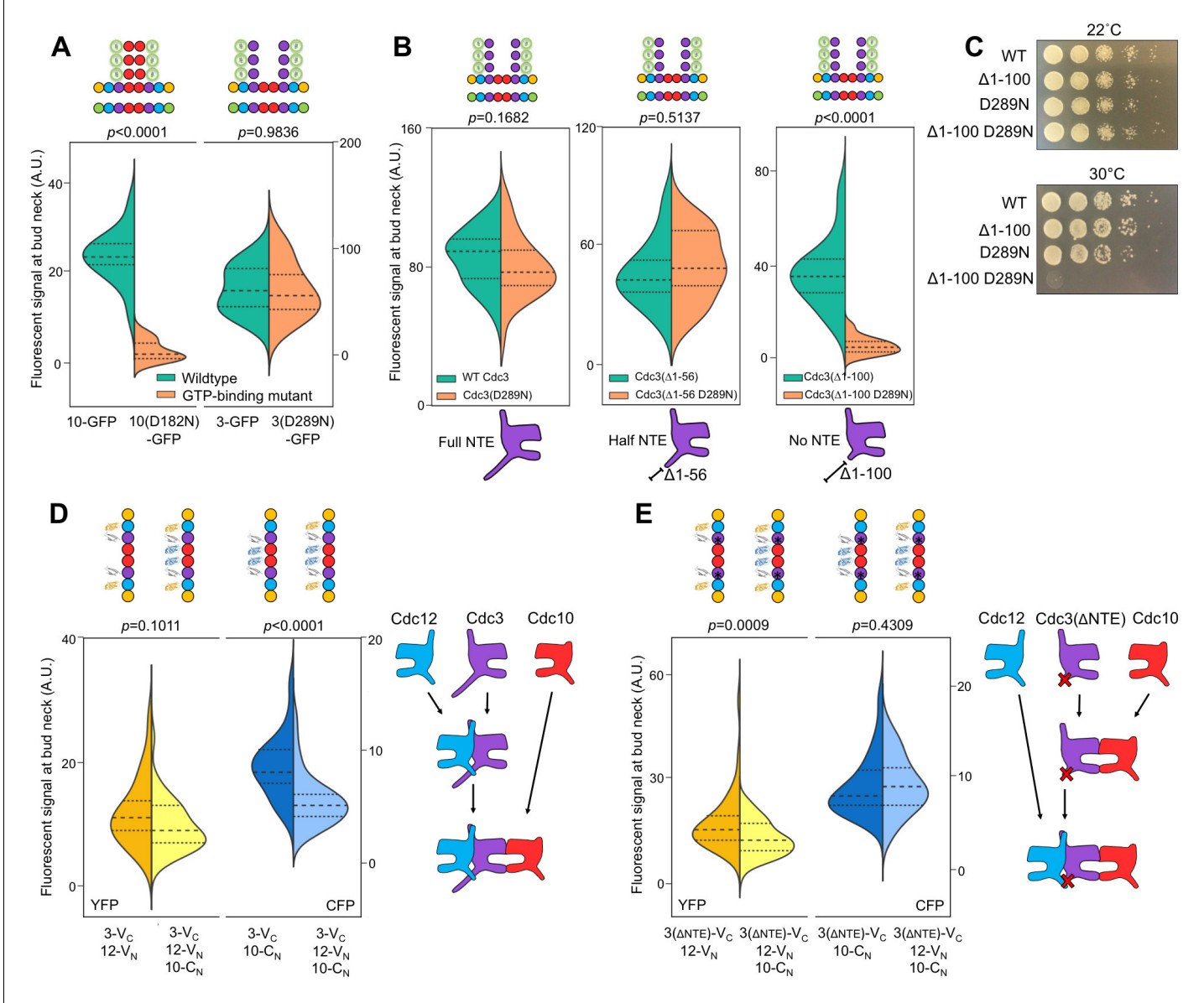

**Figure 5.** The order of Cdc3 oligomerization is controlled by the Cdc3 N-terminal extension. (A) Violin plot showing bud neck fluorescence for the indicated GFP-tagged alleles of Cdc10 or Cdc3 expressed from plasmids in WT cells (strain BY4741, plasmids pLA10, pCdc10-1-GFP, pCdc3-GFP, and YCpL-Cdc3(D289N)-GFP) cultured at 22°C to mid-log phase prior to imaging. From left to right, n = 14, 11, 14, and 10. (B) As in (A), but including plasmids (pCdc3-GFP, YCpL-Cdc3(D289N)-GFP, YCpL-Cdc3(Δ1–56)-GFP, YCpL-Cdc3(D289N, Δ1–56)-GFP, YCpL-Cdc3(Δ1100)-GFP, or YCpL-Cdc3 (D289N, Δ1–100)-GFP) encoding Cdc3-GFP alleles with D289N and/or the indicated truncations of the NTE. From left to right, n = 35, 30, 25, 16, 28, and 27. (C) Dilution series of *cdc10Δ* cells (strain JTY5104 w/ Cdc3 covering plasmid pMVB100) carrying the plasmids from (B) grown on rich medium incubated at the indicated temperature. (D–E) CSD-BiFC experiment and interpretation as in *Figure 2B*, but with different strains (YEF5692, MMY0191, or 12-V_N/10-C_N) and Cdc3-V_C or Cdc3(ΔNTE)-V_C expressed from plasmids (YCpHU-Cdc3-V_C or YCpHU-Cdc3(Δ1–100)-V_C). (D) From left to right, n = 22, 37, 43, and 33. (E) From left to right, n = 37, 66, 42, and 57.

The following figure supplements are available for figure 5:

**Figure supplement 1.** Proposed G-occlusion model of Cdc3 NTE function in septin oligomerization.

**Figure supplement 2.** Predicted disorder and conservation in the Cdc3 NTE.

**Figure supplement 3.** In silico structural modeling of G occlusion by the Cdc3 NTE.

expected consequence of bypassing the transient GTP-bound form would be a decrease in Cdc11 incorporation into septin hetero-oligomers.

MPA treatment not only depletes guanine nucleotides, it also significantly reduces the ratio of GTP:GDP in the residual nucleotide pool to nearly 1, compared to >3:1 in untreated, fermentatively growing cultures (*Saint-Marc et al., 2009*). Consistent with our prediction, MPA treatment reduced Cdc11-GFP signal in septin rings and increased Shs1-GFP signal (*Figure 6A*), mirroring with WT cells the results of mutating Cdc12 (*Figure 4B,D,F*).

We sought to confirm these findings in untreated WT cells, exploiting shifts in nucleotide ratios known to occur during normal metabolism. Reducing the concentration of fermentable carbon (dextrose) downregulates IMPDH expression (*Bradley et al., 2009*) and reduces the GTP:GDP ratio (*Rudoni et al., 2001*). IMPDH expression also decreases when dextrose is replaced with glycerol (*Roberts and Hudson, 2006*), a non-fermentable carbon source. Both 0.1% dextrose and 3% glycerol reduced Cdc11-GFP and increased Shs1-GFP incorporation into septin rings (*Figure 6C,D*). Importantly, in neither condition does the level of Shs1 or Cdc11 (as indicated by mRNA levels) change significantly (*Roberts and Hudson, 2006*; *Tai et al., 2005*). (Notably, Cdc3–Cdc12 BiFC also decreased modestly but significantly in these conditions [*Figure 6—figure supplement 1*], consistent with an overall decrease in guanine nucleotides expected to accompany IMPDH reduction, and in agreement with our findings using pharmacological IMPDH inhibition [*Figure 1*]). Taken together, these results support a model in which, via slow Cdc12 GTP hydrolysis, the process of septin assembly is sensitive to shifts in cytosolic guanine nucleotide ratios, and responds by directing distinct choices of protofilament species.

## Shs1 is required for proper septin ring assembly and function in low GTP:GDP conditions

If the increase in the Shs1:Cdc11 ratio that we observed when the cytosolic GTP:GDP ratio drops represents an adaptive response to changing cellular conditions, then cells lacking Shs1 should be particularly sensitive to such conditions with regard to septin function. When *shs1Δ* mutants of the BY4741 strain background are cultured at moderate temperatures in 2% dextrose, bud morphology and septin ring assembly (as observed by light microscopy) are largely normal (*Finnigan et al., 2015*) (*Figure 6D*). In contrast, elongated cells were frequent (~20%) in *shs1Δ* mutants cultured in low dextrose or glycerol. (~2% of *shs1Δ* cells were elongated in 2% dextrose; no elongated WT cell was found in any condition). Moreover, as visualized with fluorescently tagged Cdc3 encoded by a plasmid, misshapen septin rings were far more prevalent in *shs1Δ* cells cultured in glycerol or low dextrose (*Figure 6D*). These results support the idea that septin filaments containing Shs1 play a particularly important role in cells with a low GTP:GDP ratio, consistent with our model that specific affinity of Cdc11 for Cdc12•GTP allows cells to tailor the composition of septin filaments to meet specific cellular demands.

## The Shs1:Cdc11 ratio in septin filaments sets the diameter of 'split' septin rings

To better understand the molecular consequences of altering the Shs1:Cdc11 ratio in septin filaments, we considered known effects of Shs1 on the curvature of filamentous septin structures. In the absence of membranes, purified, recombinant hetero-octamers containing Cdc11 at both ends form straight filaments with little curvature (*Garcia et al., 2011*; *Sirajuddin et al., 2007*; *Frazier et al., 1998*; *Sadian et al., 2013*; *Bertin et al., 2008*). Titrating in Shs1-capped hetero-octamers promotes the assembly of curved filaments and, at high Shs1:Cdc11 ratios, rings of diameter ~0.3 μm (*Garcia et al., 2011*), considerably smaller than the ~1 μm diameter of septin rings in WT yeast cells (*Okada et al., 2013*). Cdc11-capped hetero-octamers associate preferentially with membrane-coated beads of diameter ~1 μm, and have little affinity for 0.3 μm beads (*Bridges et al., 2016*); Shs1-capped octamers have not been tested in this way. Based on these in vitro studies, increasing the Shs1:Cdc11 ratio might be expected to increase the curvature, and decrease the diameter, of septin rings in vivo, either by directly influencing filament curvature, or by modulating the affinity of septin filaments for regions of the plasma membrane with specific curvatures. Indeed, in *shs1Δ cdc11Δ* cells septin filaments are no longer restricted to the most highly curved regions of the bud necks (*McMurray et al., 2011*; *Iwase et al., 2007*), and in cells of the filamentous fungus *Ashbya*

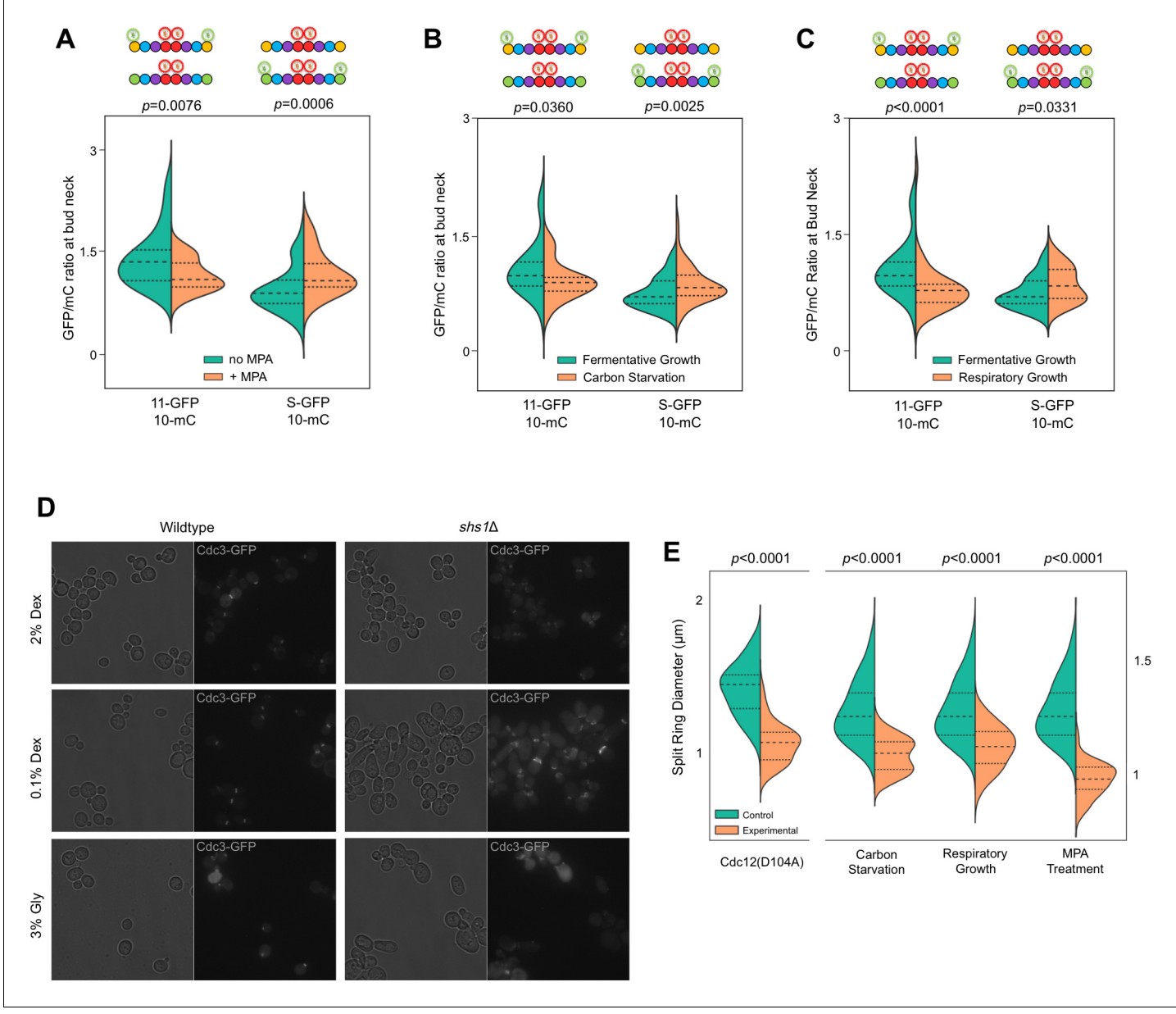

**Figure 6.** Recruitment of Cdc11 vs Shs1 by Cdc12 is sensitive to changes in the guanine nucleotide ratios. (**A–C**) Violin plots showing Cdc11-GFP-to-Cdc10-mCherry or Shs1-GFP-to-Cdc10-mCherry ratios, as in *Figure 4B*, but with cells of strain JTY5396 or JTY5397 carrying no plasmid and exposed to various physiological conditions that alter the GTP:GDP ratio. (**A**) Treatment with 100 µg/ml MPA for 24 hr. From left to right, *n* = 35, 39, 48, and 26. (**B**) Media with 2% dextrose (fermentative growth; normal) or 0.1% dextrose (carbon starvation). From left to right, *n* = 31, 39, 57, and 58. (**C**) Media with 2% dextrose (fermentative growth) or 3% glycerol (respiratory growth). From left to right, *n* = 31, 47, 57, and 26. (**D**) Micrographs showing cell morphology for WT (BY4741) or *shs1Δ* (JTY3631) cells carrying a plasmid (pML109) expressing Cdc3-GFP in fermentative, carbon starvation, or respiratory growth conditions (YP w/2% dextrose, 0.1% dextrose, or 3% glycerol, respectively). (**E**) Septin ring diameters at split-ring phase. Violin plots showing ring diameters for cells of strain JTY5396 either carrying plasmids expressing WT Cdc12 (pFM650) or Cdc12(D104A) (YCpL-Cdc12(D104A)) on the left, or grown in the described conditions (control, YP w/2% dextrose;, carbon starvation, 0.1% dextrose; respiratory growth, 3% glycerol; or 2% dextrose w/100 µg/ml MPA) on the right. From left to right, *n* = 20, 28, 25, 33, 25, 24, 25, 30.
The following figure supplement is available for figure 6:

**Figure supplement 1.** Non-pharmacological IMPDH inhibition reduces septin-septin interaction across the NC interface.

*gosyppii*, the presence of WT Shs1 similarly appears to direct septin hetero-octamer recruitment to membrane regions with distinct curvatures (*Meseroll et al., 2012*). Previous studies clearly established that the diameters of the yeast septin rings at the time of bud emergence (*Okada et al., 2013*) and the diameter of the bud neck itself (*Gladfelter et al., 2005*) are functions of the activity of the small GTPase Cdc42, and do not reflect intrinsic properties of the septin filaments themselves (although Shs1/Cdc11 composition might dictate the range of tolerable diameters). On the other hand, upon the onset of cytokinesis, the septin 'collar' at the bud neck splits into two rings flanking the hourglass-shaped bud neck (*Lippincott and Li, 1998*; *Chen et al., 2011*). We reasoned that the range of membrane curvatures available at the hourglass periphery could accommodate 'split' septin rings of various diameters, independently of the extrinsic factors that dictate the diameter of the septin collar earlier in the cell cycle.

We therefore hypothesized that effects of alterations in the Shs1:Cdc11 ratio on the curvature of septin filaments in vivo would be manifested as changes in the diameter of septin split rings. We measured split ring diameter in the same cells in which we previously measured the Shs1:Cdc11 ratio (*Figure 6A–D*). Split septin ring diameter decreased in every situation in which the Shs1:Cdc11 ratio increased (*Figure 6G*). Importantly, cells cultured in 0.1% dextrose are smaller (*Soma et al., 2014*), and, as a GTPase, Cdc42 activity may be influenced by GTP:GDP ratio changes, raising the possibility that the observed changes in septin split ring diameter were indirect effects of changes in these extrinsic factors. However, split septin rings were also smaller in *cdc12(D104A)* cells cultured in 2% dextrose (*Figure 6E*), arguing against these explanations. By EM tomography, septin filaments are grossly disorganized at the necks of *shs1Δ* cells grown in 2% dextrose, yet buds are round (*Garcia et al., 2011*). We suspect that without Shs1, septin filaments are incompatible with the tighter curvature of the plasma membrane at the bud necks of smaller cells formed in 0.1% dextrose. While further experiments will be required to more carefully investigate effects of the Shs1:Cdc11 ratio on cell size and shape, our findings provide the first hints of a functional context for the relationship that we discovered between this ratio and that of cytosolic GTP:GDP.

## Discussion

### Septin GTPase activity regulates progression through the protofilament assembly pathway and contributes to protofilament species selection

A complete nucleotide hydrolysis/exchange cycle is critical for actin and tubulin NTPases because their cellular functions rely on rapid dynamics of polymer assembly (promoted by NTP binding) and disassembly (promoted by NTP hydrolysis and release of inorganic phosphate). In these polymer systems, the irreducible building blocks ($\alpha$-$\beta$ tubulin heterodimers, or actin- or FtsZ-type monomers) must hydrolyze slowly outside the polymer context, otherwise a significant pool would, once loaded with NTP, rapidly convert to a form incompetent for polymerization. Hydrolysis rates in the polymeric context are still relatively slow for cytoskeletal NTPases compared to signaling GTPases, which is critical to allow the assembly of meta-stable polymers that persist long enough to carry out their functions.

Importantly, however, the very first GTP hydrolysis event catalyzed by a nascent tubulin polypeptide takes place concomitantly with $\alpha$-$\beta$ heterodimerization and triggers heterodimer release from a complex with tubulin folding cofactors (*Tian et al., 1999*). Allosteric communication between the septin G and NC interfaces occurring upon GTP hydrolysis, that is, NC priming, represents a conceptually similar mechanism by which an initial septin-septin interaction permits subsequent interactions with other proteins. A key, persistent assumption regarding NC priming, reminiscent of the $\alpha$-$\beta$ tubulin scenario, was that G dimerization triggers GTP hydrolysis (*Gasper et al., 2009*), and that septin monomers do not hydrolyze GTP with physiologically relevant kinetics. Our work further informs this model by providing evidence that G dimerization is not required for hydrolysis-coupled NC priming to occur. Instead, we propose that slow monomeric hydrolysis allows Cdc12 to more stably sample both GTP and GDP states prior to assembly, and thus to recruit alternate partner subunits (*Figure 7*).

By allowing formation of an interface between a GTP-bound and a GDP-bound septin (which structural studies suggest is 'tighter' than a 'GDP–GDP' interface [*Sirajuddin et al., 2007*]) loss of GTP hydrolysis by Cdc3 during septin evolution likely promoted incorporation of Cdc10 into

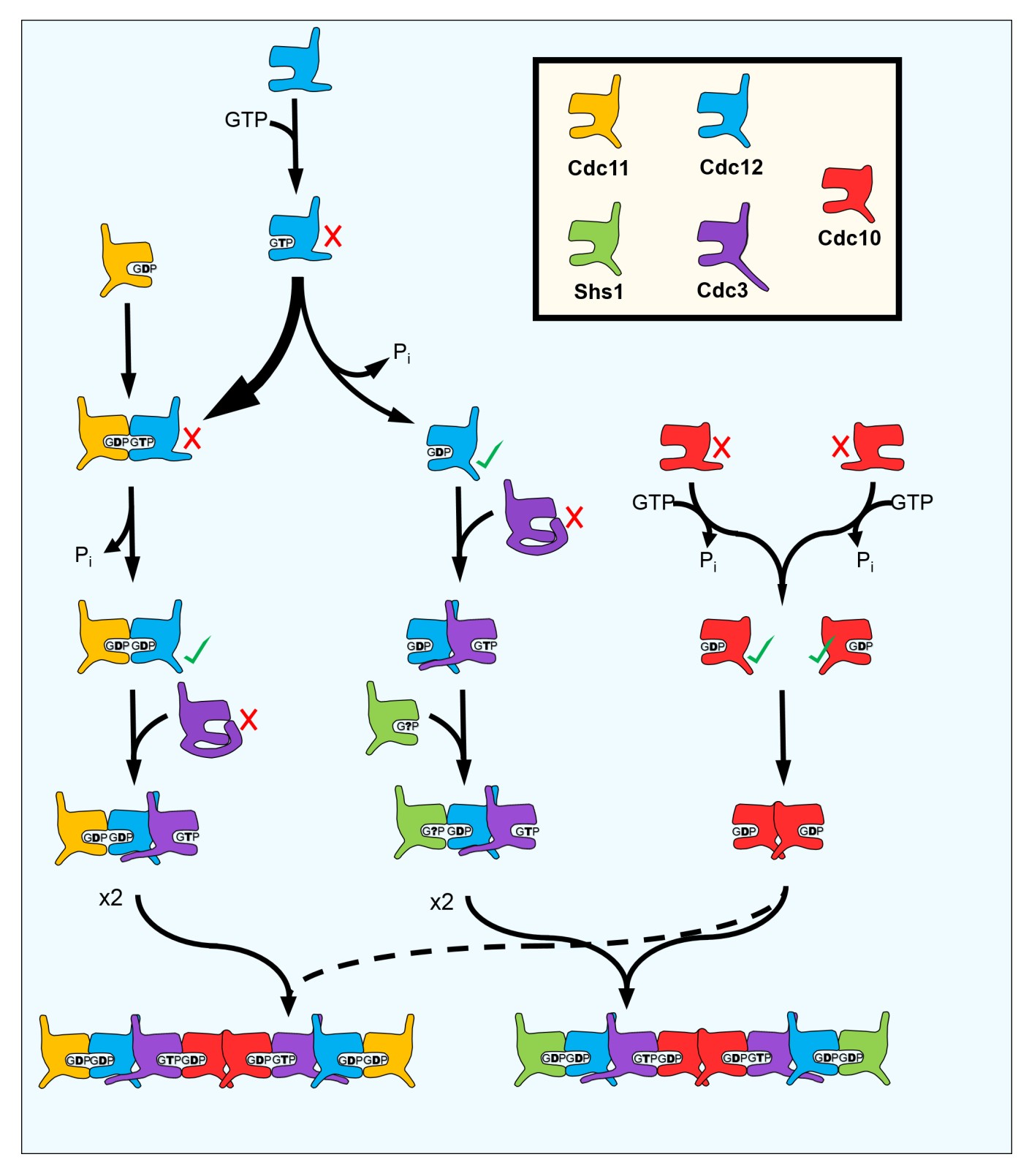

**Figure 7.** A model for the step-wise pathway of septin protofilament assembly in *S. cerevisiae*.

protofilaments, perhaps analogous to the way that 'locking' the $\alpha-\beta$ tubulin interface created an obligate heterodimer building block for microtubules. Uncoupling oligomerization from nucleotide state was also presumably important for evolution of Cdc11 and Shs1. Rounds of septin protofilament polymerization into filaments and subsequent depolymerization, manifested in vivo as septin ring assembly and disassembly during the cell division cycle, take place without significant nucleotide hydrolysis or turnover (*Vrabioiu et al., 2004*). Similarly, in vitro filament assembly and disassembly by septin hetero-octamers is controlled by the ionic strength of the buffer with no requirement for exogenous GTP (*Frazier et al., 1998*; *Bertin et al., 2008*; *Booth et al., 2015*). Hence, the NC-mediated interactions between the ends of yeast protofilaments that mediate filament polymerization (i.e. Cdc11–Cdc11 and Cdc11–Shs1, but not Shs1–Shs1 [*Booth et al., 2015*]) are not subject to control by NC priming, and extremely slow GTP hydrolysis by preassembled yeast septin protofilaments, while detectable in vitro (*Farkasovsky et al., 2005*), is likely irrelevant in vivo.

The Switch I loops wherein lie the presumptive catalytic Thr residues in Cdc10, Cdc12, and various non-yeast septins have diverged significantly in Cdc11 and Shs1 (*Garcia et al., 2011*) and individually purified Cdc11 exhibits no GTP hydrolysis in vitro (*Versele and Thorner, 2004*). Cdc11 possesses in place of Lys an unusual P-loop Arg residue whose longer sidechain is predicted to occupy the space normally filled by the $\gamma$ phosphate of GTP, making GTP binding unlikely (*Brausemann et al., 2016*). We suspect that Cdc11 binds only free GDP (*Figure 7*). Cdc11•GDP also fits both the ~1:1 septin:nucleotide ratio in purified yeast protofilaments, and a GDP:GTP ratio that is best estimated as ~3:1 by taking into account the fact that Cdc11 was substoichiometric following purification (*Frazier et al., 1998*; *Farkasovsky et al., 2005*; *Vrabioiu et al., 2004*).

## The allosteric switches that regulate yeast protofilament assembly could contribute to protofilament species selection in mammals

Counting the sporulation-specific subunits Spr3 and Spr28, yeast septins populate up to four distinct octameric protofilament species (*Garcia et al., 2011*; *Bertin et al., 2008*; *Garcia et al., 2016*). Mammalian septin protofilaments are exceedingly more complex. Indeed, with 13 septin genes – each possessing up to eight different splice variants – human cells could potentially assemble $\geq$4992 distinct protofilament species, considering only canonical palindromic octamers and hexamers. Considerable evidence of stark functional variation between septin isoforms fits with observations of distinct septin structures and protofilaments in different cell types and suggests cell-type-specific demands for different septin protofilaments (*Mostowy and Cossart, 2012*; *Dolat et al., 2014*; *Saarikangas and Barral, 2011*).

At least 7 of the 13 septin genes are transcribed in all tissues, with specific tissues expressing considerably more (*Mostowy and Cossart, 2012*). For instance, 12 of the 13 septin genes are expressed in the central nervous system (*Mostowy and Cossart, 2012*), and hippocampal neurons alone produce all 10 septins thus far queried by immunoblot (*Tsang et al., 2011*). However, not all septins produced in a given cell are incorporated into protofilaments: specific septins are excluded from hetero-oligomers containing other septins with which the excluded proteins are clearly able to interact in other contexts (*Abbey et al., 2016*; *Sellin et al., 2014*; *Mizutani et al., 2013*; *Shinoda et al., 2010*).

Given our yeast results, auto-inhibitory occlusion of G interfaces by elongated NTEs represents one possible mechanism for exclusion of specific subunits: in the absence of an NC partner competent to recruit the NTE, the G interface might remain occluded. Indeed, five human septin genes (SEPT2, SEPT3, SEPT4, SEPT9, and SEPT10) have NTEs of at least ~43 amino acids, similar to the region functional for G-occlusion in Cdc3. Each is similarly devoid of predicted motifs and consistent secondary structures. (The SEPT9 NTE interacts with actin (*Smith et al., 2015*) and microtubules (*Bai et al., 2013*), but the relevant sequences are restricted to the N-terminal half, while the C-terminal half fulfills the conditions described above.) Moreover, for SEPT2, SEPT4, SEPT9, and SEPT10, splice variants halve the NTE or eliminate it altogether, hypothetically allowing alternative splicing to control whether G-occluding or non-G-occluding variants are expressed. Mammalian septins also display wide variation in GTPase rates that could contribute to septin protofilament species selection in a manner analogous to what we propose for Cdc12–Cdc11/Shs1. Indeed, *Abbey et al., 2016* found that a SEPT7(S63A) mutation that doubles the rate of GTP hydrolysis in vitro shifts in vivo the specific SEPT9 splice variants with which it interacts (via the G interface).

We found that different metabolic states in *S. cerevisiae* cells bias the composition of septin hetero-oligomers in a manner that correlates with the cytosolic GTP:GDP ratio and GTP concentration, which we propose reflects differential availability of Cdc12•GTP during hetero-oligomer assembly. Ras GTPases are similarly sensitive to this ratio: exchange of GDP for GTP is hampered when GTP:GDP approaches 1 (*Rudoni et al., 2001*). Changes in guanine nucleotide ratios may also alter the nucleotide states adopted by monomeric mammalian septin GTPases, representing a previously unanticipated mechanism of developmental control of septin function via specification of subunit composition within septin hetero-oligomers. An increase in the GTP:GDP ratio accompanying IMP dehydrogenase induction during proliferation (*Collart et al., 1992*; *Zimmermann et al., 1998*) might drive assembly of distinct septin protofilaments in human cells. Thus, the diversity of protofilaments found in vivo may not be accurately reflected by studies with immortalized, highly proliferative cultured cells in vitro. Future studies will be required to test these models.

### Elucidating the order of hetero-oligomer assembly in vivo

Understanding at a mechanistic level the pathways by which multisubunit protein complexes are assembled is important for interpreting the effects of mutations associated with disease, among many other reasons, but this topic remains largely unexplored, in part due to technical challenges (*Marsh and Teichmann, 2015*). Electrospray mass spectrometry sensitively detects subcomplexes formed during disassembly, and relies on the assumption that assembly follows the reverse order (*Levy et al., 2008*). For complexes that assemble slowly or in discrete steps punctuated by pauses – for example, when specific subunits are synthesized only at defined points in the cell cycle – assembly intermediates may be detectable by traditional pulse-chase methods. On the other hand, mutations that block or delay specific steps are typically exploited to determine assembly pathways for complexes that, like septin hetero-oligomers, assemble rapidly without detectable WT assembly intermediates. We developed CSD-BiFC to address this shortcoming, and propose that this approach will be applicable to other complexes with equivalent properties and for which BiFC tagging does not itself perturb the assembly process. It remains to be seen for how many other complexes assembly order is, like yeast septin assembly, specific and consistent. The allosteric mechanisms we propose here facilitate assembly of hetero-oligomers with precise organization of subunits that are structurally very similar, a challenge not faced by complexes composed of subunits that, due to stark differences in shape, can only fit together in exactly one way.

## Materials and methods

### Strains, media, and genetic manipulations

All yeast strains and plasmids (*Supplementary file 1*) were manipulated using standard techniques (*Amberg et al., 2005*). Yeast cells were cultivated in liquid or on solid agar plates of rich or synthetic media, as appropriate to maintain plasmid selection. Rich growth medium was YPD (1% yeast extract (#Y20020, Research Products International Corp., Mount Prospect, IL), 2% peptone (#P20241, RPI Corp.), 2% dextrose (#G32045, RPI Corp.)). Synthetic growth medium was based on YC (0.1 g/L Arg, Leu, Lys, Thr, Trp, and uracil; 0.05 g/L Asp, His, Ile, Met, Phe, Pro, Ser, Tyr, and Val; 0.01 g/L adenine; 1.7 g/L Yeast Nitrogen Base (YNB) without amino acids or ammonium sulfate; 5 g/L ammonium sulfate; 2% dextrose) with individual components (from Sigma Aldrich, St. Louis, MO, or RPI Corp.) eliminated as appropriate for plasmid selection. For solid media, agar (#A20030, RPI Corp.) was added to 2%. G418 sulfate (Geneticin, #G1000, US Biological) was added to YPD at 200 µg/mL for selection of *kanMX*. Mycophenolic acid (MPA) was added to YPD to a final concentration of 100 µg/mL of the active drug. Sporulation was induced in 1% potassium acetate, 0.05% glucose, 20 mg/L leucine, 40 mg/L uracil. Bacterial strain DH5alpha was used to propagate plasmids. Yeast transformation was performed using the Frozen-EZ Yeast Transformation Kit II (Zymo Research).

### Preparation of yeast DNA, PCR, and cloning

Genomic DNA from yeast was isolated as described previously (*Johnson et al., 2015*). PCR was performed with various high-fidelity enzymes, typically Phusion (New England Biolabs), according to the manufacturer's instructions.

## Microscopy and image analysis

All images were captured with an EVOSfl (Advanced Microscopy Group) all-in-one microscope equipped with an Olympus 60× oil immersion objective and GFP, TXRed, YFP, and CFP filters. Yeast cells were grown to mid-log phase at room temperature on solid media, or in liquid media for MPA treatment. Cells were then washed and mounted in sterile distilled, deionized water under #1.5 coverslips. Microscope settings were individually set for each experiment to minimize exposure times and illumination intensity, with identical settings maintained for all experimental groups across each experiment. To reduce the effects of bleaching in our data, no technical replicates (i.e. imaging of the same cells twice) were performed. However, single-cell resolution allowed for a high number of biological replicates to be obtained in each experiment. Micrographs were generated as eight-bit, greyscale. tif files. Quantification of fluorescent signal was performed using the three attached ImageJ macros (*Supplementary files 2–4*) and the FIJI software suite. Briefly, the bud neck signal quantification macros operate by loading micrographs, despeckling them, and subtracting background signal using the 'rolling ball' method. The user then selects regions of interest (ROIs) around bud necks to be measured (only early bud necks [i.e. pre-collar stage] were measured in this work for the sake of consistency, and cells showing abnormal morphologies were excluded). Local maxima are then identified within these ROIs, and the user is asked to confirm or deny each maximum as an object to be measured. Upon confirmation, the pixel intensity and coordinates of the maximum are measured and the image is marked with a circle to prevent the user from erroneously measuring a single bud neck more than once. Once all maxima are processed, the data are exported as excel files. The 'matched' macro performs these same tasks, but first aligns and stacks two micrographs of the same cells imaged with different channels, so that GFP and mC signals from the same bud neck can be compared and expressed as a ratio (see *Figures 4* and *6*). Upon export to Microsoft Excel files, signals were matched using their coordinates and compiled into a single document. The 'cyto' macro performs quantification of cytosolic signal intensity by first despeckling and subtracting background as above, and then selecting small circular ROIs of uniform size (4 pixel diameter) based on user input. Mean pixel intensity within the ROIs is exported to an Excel file along with coordinates. Care was taken to avoid quantifying cytoplasmic regions comprising uncharacteristically high or low signal intensity (i.e. within the vacuole).

## Statistical analysis and kernel density estimation

Data were imported into GraphPad Prism 7.01 for statistical analysis. Data were first tested for Gaussian distributions using the D'Agostino-Pearson omnibus normality test, $\alpha = 0.05$. Groups were then compared using either two-tailed, unpaired t-tests or Mann-Whitney tests, depending on the results of the D'Agostino-Pearson test. If data were Gaussian but had different SDs according to an F-test, Welch's correction was applied to the t-test. Given a lack of expectations for effect size before the experiments were performed, no *a priori* power analyses were conducted to determine adequate *n*-values – instead, efforts were made to produce the highest *n*-values that could be reasonably achieved for each experiment. Subsequently, effect sizes were noted and power analyses were performed with this information, using point biserial correlations via G*Power 3.1.9.2 (two-tailed, $\alpha$-error prob = 0.05, 1 - $\beta$-error prob = 0.95). Importantly, these analyses used no specific values gathered from observed data (such as SD, means, etc.), but only general ranges of effect sizes to ensure sufficient *n*. All data analyzed in this way possessed adequate *n* given the effect size ranges observed for experiments of this type. KDEs and violin plots were produced using the Seaborn visualization library in the Python 2.7 IDE, Spyder 2.3.9. Kernel bandwidth was determined using Scott's rule of thumb.

## In silico structural modeling and protein-protein docking

The apo-Cdc3 structural model was generated by threading the primary sequence of Cdc3(116-411) (truncated to match the template model) through the apo-Cdc11 crystal structure (PDB 5AR1) using I-TASSER (*Yang et al., 2015*; *Roy et al., 2010*; *Zhang, 2008*). The Cdc3 α0 helix was modeled by threading its sequence through the SEPT6•GTP crystal structure published (PDB 2QAG) using I-TASSER, and attached to the apo-Cdc3 model (maintaining orientation relative to the globular domain) using Chimera 1.11. Docking was performed using a local installation of AutoDock Vina (*Trott and Olson, 2010*) via Chimera 1.11 with a search volume containing the C-terminal half of the

receptor model. The ligand was a 25-residue peptide constructed from Cdc3(57-82). Of the seven results, one was selected that positioned its C-terminal residue in a position accessible to a 18-residue linker chain (Cdc3(83-100) attached to the α0 helix. Once the linker chain and peptide ligand were attached, the linker was modelled into a low-energy conformation using MODELLER via Chimera 1.11. Ribbon diagrams were generated using Chimera 1.11 or PyMOL Molecular Graphics System, Version 1.5.0.1, Schrödinger, LLC.

## Acknowledgements

We thank Marian Farkasovsky (Slovak Academy of Sciences, Bratislava), Erfei Bi (University of Pennsylvania School of Medicine), and Jeremy Thorner (University of California, Berkeley) for strains and plasmids. We thank Colby Fees and Domenico Galati for help with ImageJ macro design; Jeff Moore, Rytis Prekeris, and Chad Pearson for critical reading of the manuscript; and Michael Polymenis (Texas A&M University) for the idea of measuring septin ring diameters. The UCSF Chimera software package is developed by the Resource for Biocomputing, Visualization, and Informatics at the University of California, San Francisco (supported by NIGMS P41-GM103311). This work was supported by the National Institute of General Medical Sciences of the National Institutes of Health under Award R00GM086603 and the Alzheimer's Association under Award NIRGD-12–241119, both to MAM.

## Additional information

### Funding

| Funder | Grant reference number | Author |
|---|---|---|
| Alzheimer's Association | NIRGD-12-241119 | Michael McMurray |
| National Institute of General Medical Sciences | R00GM086603 | Michael McMurray |

The funders had no role in study design, data collection and interpretation, or the decision to submit the work for publication.

### Author contributions

AW, Conceptualization, Data curation, Software, Formal analysis, Validation, Investigation, Methodology, Writing—original draft; MM, Conceptualization, Supervision, Funding acquisition, Methodology, Project administration, Writing—review and editing

### Author ORCIDs

Michael McMurray, http://orcid.org/0000-0002-4615-4334

## Additional files

### Supplementary files

• Supplementary file 1. Yeast strains and plasmids. Table listing the strains and plasmids used in this study, including genotypes and origins.

• Supplementary file 2. Image J macro used to analyze microscopy data for cytoplasmic fluorescence. A macro made for ImageJ to automate the quantification of cytoplasmic fluorescence.

• Supplementary file 3. Image J macro used to analyze microscopy data for cytoplasmic fluorescence. A macro made for ImageJ to automate the quantification of maximum fluorescence signal intensity in a small area (bud neck).

• Supplementary file 4. Image J macro used to analyze microscopy data for ratios of GFP and mCherry bud neck fluorescence. A macro made for ImageJ to automate the quantification of maximum fluorescence signal intensity in a small area (bud neck) matched across images of the same cells for two different filter sets.

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
