## [Decision Letter]

Thank you for submitting your article "The step-wise pathway of septin hetero-octamer assembly in budding yeast" for consideration by *eLife*. Your article has been reviewed by three peer reviewers, one of whom, Yves Barral (Reviewer #1), is a member of our Board of Reviewing Editors, and the evaluation has been overseen by Anna Akhmanova as the Senior Editor.

The reviewers have discussed the reviews with one another and the Reviewing Editor has drafted this decision to help you prepare a revised submission.

There is consensus on finding your study interesting and worth of publication in *eLife*. However, we feel that the data are not fully ready yet, and a number of additional experiments are still needed before final acceptance. The main concerns of the reviewers are two fold:

First, we feel that while your novel assay, CSD-BiFC, seems very original and powerful, it still needs to be more extensively validated before firm conclusions can be drawn. Since the vast majority of the conclusions of the paper rely only on this assay, its thorough validation is vital.

Second, your conclusions would certainly benefit from alternative support. The reviewers feel that a strong approach towards this would be if you could provide biological evidence for your conclusions and their relevance.

Essential revisions:

1) BiFC is your only readout of septin-septin interactions. To complement violin plots, provide fluorescent images to show what exactly is being imaged and quantified (presumably by ImageJ?). Please, provide experimental support for your assumption that BiFC tagging does not perturb the assembly (as stated in subsection “Elucidating the order of hetero-oligomer assembly in vivo”).

Suggestion: For example, what are the phenotypes associated with expressing the tagged proteins? Do they affect viability, the timing of cytokinesis and septin reorganization, or the morphology of the cells? Do they cause synthetic phenotypes when combined with septin mutations?

The interpretations presented in the manuscript rely on the assumption that the BiFC fragment-tagged neighboring subunits competitively interacting via G- and NC-interfaces provide equal opportunities for those fragments to assemble, such that the detected signals reflect the bias in the G- vs NC-based septin interaction. It is not clear that this is the case. Assuming the BiFC tags are always C-terminal to the septins (hard to judge, as the tagging strategy is not explicitly detailed), certain interactions may clearly be favored. The existing structural data, such as the heterotrimeric human septin assembly (pdb: 2QAG), show more than 3-fold difference in the distances of the C-termini of the septins: ~7 nm for the G-interface-related monomers, ~2 nm for the NC-interface-related ones. One would therefore expect a strong positional effect to be there. The positional bias of the BiFC signals may undermine the interpretations, and the authors should find a way to address this point. Indeed, the study relies on analysis of competing interactions of BiFC fragments, and some of these interactions may be favored over the others simply by difference in BiFC fragment proximity.

Suggestion: Using modeling, please make sure that the constructs you used, and thus also your interpretations of BiFC experiments, are consistent with the available structural evidence.

The study seems also to exclusively rely on fluorescent signal at the bud neck. How do we know BiFC signal comes from linear septin complex assembly? Considering assemblies at the bud neck are more than isolated filaments, could the readouts be confusing septin filaments versus bundles versus rings? The same positional bias may apply to the interactions of protofilaments in the context of known larger structures, such as paired septin filaments (Bertin et al., 2008, DeMay et al., 2011). How would the interpretation change if the convoluted signal from the fluorescence-based in vivo experiments were to stem from adjacent protofilaments, rather than from subunits within the individual octamers? Although you make a case for a purely in vivo study, in subsection “Depletion of guanine nucleotides perturbs both G and NC septin-septin interfaces”, it is not quite clear from the current version of the manuscript that the interactions observed in vivo can be unequivocally ascribed to individual octamers.

Suggestion: Is it possible to measure interaction between subunits in the cytoplasm, i.e., before the oligomers assemble into higher order structures at the bud neck? Can these interactions be compared to those observed at the bud neck? Perhaps isolated oligomers are easiest to observe during the G1 phase, when the old ring disassembles and the new one forms? If there are differences in results from cytoplasm versus bud neck measurements and / or a problem for other constructs used (eg signal too weak, BiFC freezing), this can also be valuable to report.

What happens to the fluorescence intensity after yeast cells divide and the septin collars disassemble? Can any biology be inferred?

2) Mycophenolic acid (MPA) is used to disrupt de novo GMP, GDP, and GTP production. Do yeast cells fail to divide in the presence of MPA? To strengthen conclusions about the role of GTP hydrolysis, can a second approach be used?

3) This report offers many new inferences including (i) the role of GTP hydrolysis, (ii) the order of septin complex assembly, and (iii) Cdc11 vs Shs1 protofilament diversity. A biological significance to confirm violin plots for any of these new inferences would strengthen the accessibility and impact of the manuscript. Some discussion on protofilmaent diversity (Shs1/Cdc11 ratio) arising from the cellular GTP/GDP ratio could be exciting and raise profile of results.

[Editors' note: further revisions were requested prior to acceptance, as described below.]

Thank you for resubmitting your work entitled "The step-wise pathway of septin hetero-octamer assembly in budding yeast" for further consideration at *eLife*. Your revised article has been favorably evaluated by Anna Akhmanova (Senior editor), a Reviewing editor and one reviewer.

We acknowledge that you have made a real effort to provide clarifications and that the manuscript has been much improved but there are some remaining issues that need to be addressed before acceptance, as outlined below:

Please, do not reserve any arguments that strengthen your manuscript. In several places you opted to exclude some text in favor of brevity (e.g., "Finally, any slight added "strength" of NC interactions due to proximity of the tags to the NC interface must not introduce a bias in the order of Cdc3-Cdc10-Cdc10 interactions, because introducing a mutation in the Cdc10 nucleotide binding pocket (D182N) randomizes the interaction order, without altering thelocation of the tags. For the sake of space, we did not add additional text to emphasize this point in the manuscript."). However, since the manuscript is not at the limit of allowed length, adding brief explanation such as this one into the text would make your paper more accessible to a broader audience.

---

## [Author Response]

*Essential revisions:*

*1) BiFC is your only readout of septin-septin interactions. To complement violin plots, provide fluorescent images to show what exactly is being imaged and quantified (presumably by ImageJ?). Please, provide experimental support for your assumption that BiFC tagging does not perturb the assembly (as stated in subsection “Elucidating the order of hetero-oligomer assembly in vivo”).*

*Suggestion: For example, what are the phenotypes associated with expressing the tagged proteins? Do they affect viability, the timing of cytokinesis and septin reorganization, or the morphology of the cells? Do they cause synthetic phenotypes when combined with septin mutations?*

We now provide, in Figure 1—figure supplement 2, fluorescence micrographs to demonstrate more clearly what kinds of cells we imaged and how the quantification of fluorescence was performed. We have also provided the FIJI/Image J scripts/macros that we created for this analysis. As experimental support of the assumption that the BiFC tagging does not perturb the assembly process in any fundamental way, we also provide, in Figure 1—figure supplement 3, new data showing that the tagging does not detectably affect viability/proliferation rate, or the timing of major structural changes in the septins associated with the onset of budding, cytokinesis, etc. (Because the reconstituted Venus fluorophore photobleaches quickly, we were unable to perform time-lapse imaging of single cells, and instead compared the frequencies of cells in different stages as visualized with snapshots of asynchronous cultures.) We have added additional body text (subsection “Depletion of guanine nucleotides perturbs both G and NC septin-septin interfaces”, paragraph three) to point out that elongated buds (a known phenotype of tagging septins) are very rare, and that we eliminated such cells from our analysis. As suggested, we also observed no obvious synthetic effect of combining CSD-BiFC tags with septin mutations, specifically the cdc10(D182N) and cdc3(Δ1-100) alleles. Thus we are confident that BiFC tagging itself does not perturb septin assembly in any way that is relevant to our conclusions.

*The interpretations presented in the manuscript rely on the assumption that the BiFC fragment-tagged neighboring subunits competitively interacting via G- and NC-interfaces provide equal opportunities for those fragments to assemble, such that the detected signals reflect the bias in the G- vs NC-based septin interaction. It is not clear that this is the case. Assuming the BiFC tags are always C-terminal to the septins (hard to judge, as the tagging strategy is not explicitly detailed), certain interactions may clearly be favored. The existing structural data, such as the heterotrimeric human septin assembly (pdb: 2QAG), show more than 3-fold difference in the distances of the C-termini of the septins: ~7 nm for the G-interface-related monomers, ~2 nm for the NC-interface-related ones. One would therefore expect a strong positional effect to be there. The positional bias of the BiFC signals may undermine the interpretations, and the authors should find a way to address this point. Indeed, the study relies on analysis of competing interactions of BiFC fragments, and some of these interactions may be favored over the others simply by difference in BiFC fragment proximity.*

*Suggestion: Using modeling, please make sure that the constructs you used, and thus also your interpretations of BiFC experiments, are consistent with the available structural evidence.*

We now specify in the body text (subsection “Depletion of guanine nucleotides perturbs both G and NC septin-septin interfaces”) that all BiFC tags are C terminal. We added other new text to emphasize that interactions between septin C termini must not drive septin-septin BiFC, or we would not be able to see differences upon nucleotide depletion, and to cite published evidence that Cdc3–Cdc12 CTE interactions are insufficient for incorporation into septin filaments/bud neck localization. Similarly, we added new text (subsection “Discrete steps in septin protofilament assembly”) pointing out that CTE-mediated interactions between Cdc3 and Cdc12 must not precede, and thus cannot “drive”, Cdc3–Cdc12 interactions, otherwise there would be no difference in order between Cdc3–Cdc12–Cdc11 and Cdc3–Cdc12–Shs1. We have added new data (Figure 2—figure supplement 3) to demonstrate that BiFC signals from C-terminally tagged CTE-containing septins are of similar “strength” regardless of whether the interaction measured is via the G or NC interface. We have also added to the legend of Figure 2—figure supplement 3 text citing published work that septin CTEs are so long and flexible (≥10 nm, rotating ~180°) that in fact the available structural evidence predicts no bias introduced by the locations of the tags, which fits with our findings. Finally, any slight added “strength” of NC interactions due to proximity of the tags to the NC interface must not introduce a bias in the order of Cdc3–Cdc10–Cdc10 interactions, because introducing a mutation in the Cdc10 nucleotide binding pocket (D182N) randomizes the interaction order, without altering the location of the tags. For the sake of space, we did not add additional text to emphasize this point in the manuscript.

*The study seems also to exclusively rely on fluorescent signal at the bud neck. How do we know BiFC signal comes from linear septin complex assembly? Considering assemblies at the bud neck are more than isolated filaments, could the readouts be confusing septin filaments versus bundles versus rings? The same positional bias may apply to the interactions of protofilaments in the context of known larger structures, such as paired septin filaments (Bertin et al., 2008, DeMay et al., 2011). How would the interpretation change if the convoluted signal from the fluorescence-based in vivo experiments were to stem from adjacent protofilaments, rather than from subunits within the individual octamers? Although you make a case for a purely in vivo study, in subsection “Depletion of guanine nucleotides perturbs both G and NC septin-septin interfaces”, it is not quite clear from the current version of the manuscript that the interactions observed in vivo can be unequivocally ascribed to individual octamers.*

*Suggestion: Is it possible to measure interaction between subunits in the cytoplasm, i.e., before the oligomers assemble into higher order structures at the bud neck? Can these interactions be compared to those observed at the bud neck? Perhaps isolated oligomers are easiest to observe during the G1 phase, when the old ring disassembles and the new one forms? If there are differences in results from cytoplasm versus bud neck measurements and / or a problem for other constructs used (eg signal too weak, BiFC freezing), this can also be valuable to report.*

*What happens to the fluorescence intensity after yeast cells divide and the septin collars disassemble? Can any biology be inferred?*

If septin-septin CSD-BiFC exclusively, or even primarily, reports on interactions between pre-formed protofilaments that occur during polymerization on the membrane, then this approach would fail miserably, at least according to available data for how protofilaments encounter one another during polymerization (see PMID: 24469790). We would expect mutual depletion in every case, i.e., random order. Furthermore, there is no current model for such post-assembly events that could explain how MPA treatment and specific point mutations within individual septins (Cdc10(D182N), Cdc3(Δ1-100)) could bring about the specific changes in CSD-BiFC signal that we observed. Even if every possible BiFC event occurs in the cytosol during assembly and prior to interaction with the membrane at the site of budding, we do anticipate some frequency of postassembly BiFC events, such as upon “lateral” association between a protofilament containing only VN-tagged septins and protofilament containing only VC-tagged septins. Such events should “crosslink” protofilaments together, which, if they occurred frequently, might be expected to perturb septin ring disassembly or the next round of septin ring assembly (pre-existing septin complexes are recycled for use through many successive cell divisions). We found no evidence for such perturbations in the strains that we used to draw our conclusions (see above). However, we now provide new data (Figure 1—figure supplement 3) for a scenario in which BiFC should induce protofilament-protofilament crosslinking upon events occurring at the bud neck, i.e., Cdc11-VC–Cdc11-VN BiFC. Here, we do observe perturbations in the timing of higherorder structural transitions: the patch-to-ring transition during late G1-early S phase is slightly prolonged. Since targeted exocytosis is thought to drive the patch-to-ring transition by delivering septin-free membrane in the center of the septin patch (PMID: 23906065), it may be the case that end-to-end-crosslinked protofilaments are less easily displaced by exocytic delivery than are “free” protofilaments, slowing clearance from the center of the patch. We mention this possibility in the legend to Figure 1—figure supplement 3, but we believe that this idea is too speculative to warrant description in the body text. We were able to quantify cytosolic BiFC signal in the nucleotide depletion experiments (Figure 1), and the trends were exactly the same as we measured at the bud neck. However, CSD-BiFC signals are, by definition, weaker, and we were unable to reliably distinguish cytosolic CSD-BiFC signal from background. We have added new text to that effect. With regard to measuring cells in G1 during the period between old ring disassembly and new ring assembly, in rapidly dividing cells like the ones we imaged (and which we believe are probably most representative of the conditions under which septins assembly pathways evolved), new ring assembly is usually concomitant with old ring disassembly (see PMID: 21736496), hence the requested cells are too rare for reliable numbers. Ultimately, given the slow maturation of the reconstituted fluorophore following a BiFC event (~50 min; PMID: 18573091), compounded with septin stability and “recycling” through cell division, in every such cell the signal reports primarily on events that occurred during previous cell cycles. Rather than detailing all of these lines of logic in the body text, if this manuscript is published in *eLife*, we hope that a curious reader could find them in the “Author Response” section.

*2) Mycophenolic acid (MPA) is used to disrupt* de novo *GMP, GDP, and GTP production. Do yeast cells fail to divide in the presence of MPA? To strengthen conclusions about the role of GTP hydrolysis, can a second approach be used?*

We have revised the body text to point out that cells treated with this concentration of MPA continue to divide. We also now include new data (Figure 6—figure supplement 1) demonstrating that NC interaction between Cdc3 and Cdc12 is also perturbed in culture conditions (reduced dextrose, or in glycerol) in which IMDH expression is reduced, and thus guanine nucleotide levels should be depleted.

*3) This report offers many new inferences including (i) the role of GTP hydrolysis, (ii) the order of septin complex assembly, and (iii) Cdc11 vs Shs1 protofilament diversity. A biological significance to confirm violin plots for any of these new inferences would strengthen the accessibility and impact of the manuscript. Some discussion on protofilmaent diversity (Shs1/Cdc11 ratio) arising from the cellular GTP/GDP ratio could be exciting and raise profile of results.*

We now include new data (Figure 6) demonstrating that in conditions in which

protofilament composition is biased towards incorporation of Shs1, the diameter of split septin rings decreases, consistent with previous in vitro studies demonstrating that increasing the Shs1:Cdc11 ratio increases the curvature of ring structures formed from mixtures of Cdc11- and Shs1-containing protofilaments. Furthermore, we now include other new data (Figure 6) demonstrating that when Shs1 is unavailable for incorporation (i.e., in *shs1Δ* cells), culture conditions that normally promote increased incorporation of Shs1 exacerbate the defects in morphology of both the cells and of septin rings. While we are careful not to suggest that the composition of septin protofilaments directly dictates the diameter of the septin collar or the bud neck, and we acknowledge that we do not yet fully understand the functional consequences of altering split septin ring diameter, we believe these new findings provide the kind of biological significance that was suggested might raise the profile of the study.

[Editors' note: further revisions were requested prior to acceptance, as described below.]

*We acknowledge that you have made a real effort to provide clarifications and that the manuscript has been much improved but there are some remaining issues that need to be addressed before acceptance, as outlined below:*

*Please, do not reserve any arguments that strengthen your manuscript. In several places you opted to exclude some text in favor of brevity (e.g., "Finally, any slight added "strength" of NC interactions due to proximity of the tags to the NC interface must not introduce a bias in the order of Cdc3-Cdc10-Cdc10 interactions, because introducing a mutation in the Cdc10 nucleotide binding pocket (D182N) randomizes the interaction order, without altering thelocation of the tags. For the sake of space, we did not add additional text to emphasize this point in the manuscript."). However, since the manuscript is not at the limit of allowed length, adding brief explanation such as this one into the text would make your paper more accessible to a broader audience.*

To the section of body text following the initial CSD-BiFC results with wild-type septins, we added the following text, which is a condensed version of arguments we made previously in our response to reviews:

“We note that, given the slow maturation of the reconstituted fluorophore following a BiFC event (~50 min (Kerppola, 2008)), compounded with long septin half-lives and septin “recycling” through cell divisions (McMurray and Thorner, 2008), the BiFC signals we measured reported primarily on interactions that occurred during previous cell cycles. However, if even a significant fraction of septin-septin CSD-BiFC signals reflected associations between pre-formed protofilaments that occurred during polymerization at the plasma membrane, then we would expect mutual depletion in every case (i.e., apparently random order), at least according to available data for how protofilaments encounter one another during polymerization (Bridges et al., 2014).”

As specifically requested, we added to the appropriate section (subsection “When Cdc10 cannot bnd nucleotide, Cdc10 homodimerization no longer precedes heterodimerization with Cdc3”) a modified version of the text regarding the ability of Cdc10(D182N) to disrupt interaction order without altering location of the BiFC tags: “Notably, disruption of interaction order by a substitution in the Cdc10 NBP demonstrates that any added NC interface affinity due to the proximity of C-terminal BiFC tags does not introduce a bias in interaction order.”